# Simultaneous Multi-objective Alignment Across Verifiable and Non-verifiable Rewards

## Abstract

Aligning large language models to human preferences is inherently multidimensional, yet most pipelines collapse heterogeneous signals into a single optimizeable objective. We seek to answer what it would take to simultaneously align a model across various domains spanning those with: verifiable rewards (mathematical accuracy), non-verifiable subjective preferences (human values), and complex interactive scenarios (multi-turn AI tutoring dialogues). Such multi-objective reinforcement learning setups are often plagued by the individual objectives being at odds with each other, resulting in inefficient training and little user control during inference. We propose a unified framework that: (i) standardizes process reward model (PRM) training across both verifiable and non-verifiable settings to better supervise models' chain-of-thought reasoning; (ii) performs multi-objective alignment by training the LLM with our **M**ulti-**A**ction-**H**ead **DPO** (MAH-DPO) and a vectorized reward where the dimensions of the vector correspond to the various objectives instead of a single scalar; and (iii) demonstrates how such a system provides fine-grained inference-time user control. Experiments across math reasoning, value alignment, and multi-turn dialogue show that our framework improves performance across multiple objectives simultaneously, while minimizing cross-objective trade-offs and enabling flexible inference time user control.

## 1 Introduction

The success and widespread deployment of large language models (LLMs) have created opportunities for AI assistance across diverse applications, ranging from mathematical problem solving and question answering to educational tutoring (7; 45; 38; 20; 44; 21). However, these real-world applications often demand that models simultaneously satisfy multiple objectives, which exposes a fundamental challenge that aligning LLMs to human preferences is inherently multidimensional (3; 4; 31). For instance, a question-answering system should provide helpful responses while being harmless (18; 48), and an AI education tutor must be able to guide students toward accurate understanding while remaining pedagogically engaging (42; 46). These scenarios span three distinct categories of alignment targets: domains with verifiable rewards where correctness can be automatically checked (e.g., mathematical accuracy), domains with non-verifiable subjective preferences that require human judgment (e.g., helpfulness, honesty, truthfulness), and complex interactive scenarios involving multi-turn dialogues (e.g., AI tutoring engagingness) where success depends on the downstream impact of the assistant's responses on subsequent user behavior.

Current alignment methods struggle to capture multi-dimensional human preferences. Common practices such as reinforcement learning from human feedback (RLHF) (11; 45) distill human comparisons into scalar reward scores for maximizing expected reward. While direct preference optimization (DPO) (49) eliminates the reward model, it still optimizes along a single preference axis. Both approaches collapse rich, structured human feedback into one-dimensional training signals, discarding valuable trade-off information and resulting in mismatches between nuanced human preferences and simplified optimization objectives.

Several recent works address multi-objective RLHF alignment through linear scalarization (33; 26; 76; 62; 19) or post-hoc parameter merging of specialized models (50; 29). However, these approaches are computationally expensive and typically require retraining when incorporating additional objectives or altering the balance among existing ones. More computationally lightweight

methods like MODPO (76) extend DPO to multiple objectives but they still apply fixed dimensional weights during training time, limiting alignment flexibility as the dimension weights cannot be changed at inference time. Alternative test-time alignment methods use reward models to guide generation step-by-step but suffer from granularity mismatches between between reward definition and generation decisions (30; 14). For example, outcome reward models are trained to score complete responses while step-level guided decoding operates on partial and incomplete responses, resulting in inconsistencies (32; 67). Recent approaches attempt to address these granularity issues by using more granular reward signals from process reward models (36; 59; 40; 25; 66). However, these solutions mostly focus on verifiable domains where intermediate steps can be reliably evaluated (74; 75) and training PRMs in non-verifiable domains remain a challenge.

To address these limitations, we develop a framework that handles multi-objective alignment through three coordinated components. **First**, we standardize PRM training across both verifiable and non-verifiable settings to enable reliable step-level supervision across domains. For verifiable domains, we augment Monte Carlo rollouts with hindsight credit assignment for reward collection. For non-verifiable domains, we devise three reward labeling strategies: i) majority voting evaluation, ii) direct step judgment, and iii) step reward approximation, based on the process definition and rollout difficulty in specific tasks. **Second**, we introduce **M**ulti-**A**ction-**H**ead **DPO** (MAH-DPO) to preserve the multi-dimensional nature of human preferences during training. Specifically, we first employ specialized action heads on top of a shared LLM backbone, where each head corresponds to a preference dimension. Then simultaneously, each head is optimized with its dimensional-specific DPO loss while the shared base LLM is updated with the cross-dimension gradient. Thus, our MAH-DPO reduces cross-objectives gradient interference for more stable training and the multi-head design also enables more flexible adaptation during inference. **Finally**, we complement training-time optimization with our PRM-guided decoding with continuing hidden-state, which offers more fine-grained user control over different objectives as well as improved alignment performances with preserved generation continuity. Together these components turn multi-objective alignment into a coherent training and inference procedure that generalizes across verifiable and non-verifiable domains with flexibilities for controllable inference-time search across each preference dimension. To summarize, we make the following contributions:

- We develop a standardized PRM training pipeline that systematically addresses the challenge of deriving fine-grained supervision across verifiable and non-verifiable domains.
- We propose vectorized multi-objective alignment via Multi-Action-Head DPO, which preserves the multi-dimensional structure of human preferences during training and enables fine-grained preference dimension control during inference.
- Extensive experiments across math reasoning, human value alignment, and multi-turn AI tutoring demonstrates the effectiveness of our multi-objective alignment framework in both training-time and inference-time optimization with possible synergy.

## 2 RELATED WORK

**Process Reward Model.** Process supervision addresses a core limitation of outcome-only evaluation by giving rewards on intermediate reasoning steps, helping systems avoid trajectories that look correct but contain logical errors. The foundational approach involves collecting step-level human annotations for mathematical reasoning tasks and training process reward models on these dense supervision signals (36; 63). Follow-up work scales supervision with automated or weakly supervised labels, for example per-step Monte Carlo rollouts or self-generated labels (59; 40). Beyond standard PRMs, recent variants introduce progress or verifier signals that score both partial correctness and future success, improving search and ranking during decoding (10; 51). There are also training objectives that regularize PRMs to improve stability (72). Practical studies discuss data generation, evaluation pitfalls, and how PRMs differ from value functions that predict eventual solvability from partial traces (74). Process-level search with step-wise scoring has further been shown to beat outcome-level test-time compute baselines in several setups, including controlled decoding, tree-structured search, and value/verification-guided search (43; 39; 70; 54; 51; 58).

**Multi-Objective Alignment.** Multi-objective alignment trains or steers language models for multiple, potentially conflicting objectives such as helpfulness, harmlessness, and honesty (65). Standard RLHF pipelines fit a scalar reward and fine-tune with PPO, or use scalarized preference optimization

(45; 49; 71; 64; 16), but they collapse trade-offs into one score. Two lines of work relax this restriction. Training-time methods adapt ideas such as multi-objective RLHF, multi-objective preference optimization, or parameter mixing to balance different rewards (76; 50; 57; 68; 52; 34). Complementing these training-based methods, test-time alignment enables dynamic objective balancing without retraining. These approaches modify token probability distributions using reward guidance and perform search under composite objectives, achieving improvements on preference benchmarks while supporting per-user customization (30; 9; 69; 37). This paradigm offers particular promise for multi-objective alignment where individual user preferences vary significantly.

# 3 BACKGROUND

To understand the challenges and opportunities in multi-objective alignment, we examine three representative domains. **Mathematics.** Mathematics represents a typical verifiable domain where ground truth can be automatically determined with datasets such as GSM8K (12), MATH (24), GaoKao (73), and OlympiadBench (23). The verifiable nature of mathematical correctness enables automatic reward assignment at both outcome and process levels. Recent work has demonstrated the effectiveness of process reward models (36; 59; 56) that provide step-by-step supervision to validate intermediate reasoning steps. Mathematical problem-solving can also involve dimensions beyond accuracy, including explanation clarity for diverse user expertise levels and pedagogical engagement in practical applications. **Human Values.** Unlike mathematical correctness, human values include a broad range of subjective preferences that cannot be automatically verified, including aspects such as helpfulness, harmlessness, and honesty (3; 4; 45; 5). These qualities require human judgment and are subjective, context-dependent, and sometimes conflicting. Recent work such as HelpSteer (61; 60) and UltraFeedback (13) provides multi-dimensional annotations and reference comparisons across multiple criteria including helpfulness, coherence, and truthfulness. The challenge lies in the subjectivity and multi-dimensionality of human preferences, while the lack of automatic verification makes it difficult to provide more fine-grained supervision. **Interactive AI Tutoring.** Interactive AI tutoring represents another challenging domain that combines objective and subjective evaluation within multi-turn dialogues, where success depends not only on correctness but also on pedagogical effectiveness, engagement, and scaffolding strategies. Datasets in this domain include educational dialogue corpora (55; 41; 8) and socratic questioning collections (53; 2; 15). Unlike static domains, the quality of a tutor's response should be evaluated based on its impact on subsequent student responses and learning trajectories. We provide an example AI tutoring dialogue in Appendix I.

# 4 PROCESS REWARD MODEL TRAINING

With varying degrees of verifiability and supervision granularity, we first develop a standardized PRM training framework across domains to lay foundations for multi-objective alignment.

## 4.1 VERIFIABLE DOMAINS

For tasks with objective correctness criteria, e.g., math, we augment the step-level supervision with outcome signals with a value target estimator to train PRMs that both validate current intermediate step and predict future correctness.

**Step-level Reward.** Given a trajectory $y_{1:N} = (y_1, y_2, \ldots, y_N)$, the step-level reward is defined as a correctness signal that captures both textual validity and local logical coherence at step $y_t$ (36; 40). Common practice of obtaining process reward labels involves a multi stage sampling and annotation process (59; 36; 66). For example, in Math Shepherd (59), multiple completions are sampled from each intermediate step to the final answer. A step is labeled as correct if at least one completion leads to a correct final solution, and incorrect if all completions result in wrong answers.

**Value Reward with Hindsight Relabeling.** Motivated by experience replay in reinforcement learning (1; 22), we perform hindsight relabeling in addition to the step-level reward. From each step $y_t$, we rollout to its completion $y_{t+1:} = (y_{t+1}, \ldots, y_n)$ and evaluate the final solution to obtain a binary terminal correctness reward $z \in \{0, 1\}$. Then, we collect a step-level reward $r_t$ from annotation or existing PRM's judgment for step $y_t$ and blend it with the discounted terminal reward to credit the current step's contribution to the final outcome as $\tilde{r}_t$. For each step $y_t$, we generate $M$ independent rollouts and aggregate them to obtain the final value target $V_t^{\text{target}}$, which is used to train the PRM by

minimizing the mean squared error on its predictions $p_t$:

$$\tilde{r}_t = r_t + \gamma^{n-t}z, \qquad V_t^{\text{target}} = \frac{1}{M}\sum_{m=1}^{M}\tilde{r}_t^m, \qquad \mathcal{L}_{\text{PRM}} = \mathbb{E}_{t,y_{1:t}}\left[\left(p_t - V_t^{\text{target}}\right)^2\right], \qquad (1)$$

where $\gamma \in (0,1)$ is a discount factor that assign credits based on temporal distance. The relabeled reward enables the PRM to predict both local step reasoning quality and future solution correctness.

### 4.2 Non-verifiable Domains

For domains lacking objective correctness measures, we adapt our PRM training framework based on the availability of clear process structure and rollout difficulty.

**Case A: Clear Process Structure with Efficient Rollout.** When the task has clearly-defined intermediate steps that can be meaningfully evaluated, e.g. engagement in math reasoning process, we employ a rollout-based labeling strategy similarly to our verifiable domain approach. We first calibrate an LLM-as-Judge $J$ using a few human annotated ratings $\hat{R}$ to approximate the expected human judgment, $J(y_{1:t}) \approx \mathbb{E}[\hat{R}]$. Then we sample $M$ completions from each step $y_t$ and evaluate the resulting full trajectories using our calibrated LLM-as-Judge $J$. We label the step $y_t$ as positive when the majority of completions are judged as positive by $J$:

$$r_t = \mathbf{1}\left[\left(\frac{1}{M}\sum_{m=1}^{M}\mathbf{1}_{\text{positive}}[(J(y_{1:t},y_{t+1:n}^m)]\right) > \tfrac{1}{2}\right]. \qquad (2)$$

This majority voting criterion reflects the inherent subjectivity in non-verifiable domains, where a reasoning step's quality is measured by its tendency to lead to generally acceptable outcomes rather than definite correctness.

**Case B: Clear Process Structure with Costly Rollout.** When generating rollouts is costly or difficult, for example multi-turn dialogue which requires real user interactions, we directly query the LLM-as-Judge $J$ on observed trajectory prefixes to otain the training label: $r_t = J(y_{1:t})$. This approach trades the robustness of rollout-based evaluation for computational efficiency. One can mitigate the increased label noise inherent in this approach through careful judge calibration, ensemble methods, and multi annotator agreement when feasible.

**Case C: Unclear Process Structure.** For domains where step wise decomposition lacks clear structures, for example general question answering tasks, we approximate the process modeling through directly evaluating the partial response with a reward model trained with complete responses. For example, one may collect or reuse available pairwise preference data $\{(y^w, y^l)\}$ to train a Bradley-Terry model (6) to score the process generation $R_\phi(y_{1:t}) \to \mathbb{R}$. The trained reward model provides holistic quality assessment that serves as guidance during decoding, approximating the intermediate process supervision even when the process structure is not well defined.

## 5 Alignment: Training and Decoding

To align LLMs for multiple objectives across domains, we propose our Multi-Action-Head DPO (MAH-DPO) for training time optimization (Section 5.1) and utilize our trained PRM directly for test-time alignment with reward-guided decoding with continuing hidden-state (Secrtion 5.2).

### 5.1 Training-Time Optimization: Multi-Action-Head DPO

**Direct Preference Optimization.** DPO (49) optimizes a policy $\pi_\theta$ against a fixed reference policy $\pi_{\text{ref}}$ using preference pairs $\mathcal{D} = \{(x, y^w, y^l)\}$, where $y^w$ is the preferred response to prompt $x$ and $y^l$ is the dispreferred one. The DPO loss is:

$$\mathcal{L}_{\text{DPO}}(\pi_\theta; \pi_{\text{ref}}) = -\mathbb{E}_{(x,y^w,y^l)\sim\mathcal{D}}\left[\log\sigma\left(\beta\left(\log\frac{\pi_\theta(y^w\mid x)}{\pi_{\text{ref}}(y^w\mid x)} - \log\frac{\pi_\theta(y^l\mid x)}{\pi_{\text{ref}}(y^l\mid x)}\right)\right)\right], \qquad (3)$$

where $\sigma(\cdot)$ is the sigmoid and $\beta > 0$ is a parameter controlling the strength of the preference signal.

**Multi-Action-Head LLM.** To jointly optimize for $H$ distinct objectives while maintaining computational efficiency, we propose the multi-action-head LLM that extends the base LLM with specialized

output layers. We maintain a single shared LLM backbone $\theta_b$, while introducing $H$ distinct linear projection heads, one for each alignment objective. This is more efficient than training $H$ separate models, which would require $H$ times the computational resources and fail to leverage cross objective synergies. Specifically, let $h_{\theta_b}(x, y_{1:t}) \in \mathbb{R}^d$ denote the $d$-dimensional hidden state produced by the shared LLM backbone $\theta_b$ for input prefix $(x, y_{1:t})$. Each objective $i \in \{1, \ldots, H\}$ has a dedicated projection head parameterized by matrix $W_i \in \mathbb{R}^{d \times |V|}$ to produce objective-specific logits $z_i$ and token probability distribution:

$$z_i(x, y_{1:t}) = W_i^\top h_{\theta_b}(x, y_{1:t}), \qquad \pi_{\theta_b, W_i}(y_t \mid x, y_{1:t}) = \text{softmax}(z_i(x, y_{1:t})) \tag{4}$$

where $|V|$ is the vocabulary size. The shared LLM backbone captures general language understanding and generation capabilities, while specialized heads can encode objective-specific preferences. During inference, our multi-action-head architecture supports flexible objective control by either selecting a specific head $i$ for targeted behavior or ensembling logits from multiple heads for balanced performance:

$$z_{\text{mix}} = \sum_{i=1}^{H} w_i z_i, \quad \pi_{\text{MAH}}(y_t \mid x, y_{1:t}) = \text{softmax}(z_{\text{mix}}). \tag{5}$$

where $w_i \geq 0$ are ensemble weights with $\sum_i w_i = 1$. This flexibility enables the model to be adapted for different downstream applications and user preferences without requiring separate training runs for each objective combination.

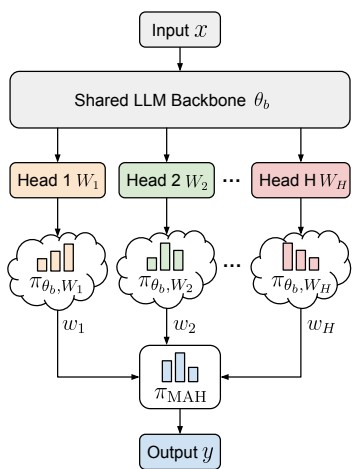

Figure 1: Overview of Multi-Action Head LLM.

**Multi-Action-Head DPO Objective.** We first curate $H$ preference datasets $\{\mathcal{D}_i\}_{i=1}^H$, where each $\mathcal{D}_i$ contains preference pairs specifically designed for objective $i$ labeled using our trained PRM or from annotated labels. All heads $W_i$ are initialized from the same language modeling head from the supervised fine-tuned (SFT) LLM $\pi_{\theta_b}$ with small random perturbations to encourage specialization. The reference model $\pi_{\text{ref}}$ retains a frozen copy of the base LLM backbone and the unperturbed SFT head which do not share any parameter with the trainable policy model. During training, examples $(x, y^w, y^l) \in \mathcal{D}_i$ are routed to head $i$, and we compute the objective-specific DPO loss:

$$\mathcal{L}_i(\theta_b, W_i) = -\mathbb{E}_{(x, y^w, y^l) \sim \mathcal{D}_i} \left[ \log \sigma \left( \beta \left( \log \frac{\pi_{\theta_b, W_i}(y^w \mid x)}{\pi_{\text{ref}}(y^w \mid x)} - \log \frac{\pi_{\theta_b, W_i}(y^l \mid x)}{\pi_{\text{ref}}(y^l \mid x)} \right) \right) \right]. \tag{6}$$

Let a mini-batch during training be partitioned as $\mathcal{B} = \bigsqcup_{i=1}^H \mathcal{B}_i$ where $\mathcal{B}_i$ gathers the examples assigned to head $i$. The combined loss we are minimizing is

$$\mathcal{L}_{\text{MAH-DPO}}(\theta_b, \{W_i\}) = \sum_{i=1}^H \alpha_i \cdot \frac{1}{|\mathcal{B}_i|} \sum_{(x, y^w, y^l) \in \mathcal{B}_i} \mathcal{L}_i(\theta_b, W_i; x, y^w, y^l), \tag{7}$$

where $\alpha_i \geq 0$ are objective weights with $\sum_i \alpha_i = 1$.

**Gradient Analysis.** The gradients for parameters of each head $j$ are isolated by routing, while the backbone LLM gradients accumulate across heads:

$$\nabla_{W_j} \mathcal{L} = \sum_{i=1}^H \alpha_i \cdot \frac{1}{|\mathcal{B}_i|} \sum_{(x, y^w, y^l) \in \mathcal{B}_i} \underbrace{\nabla_{W_j} \mathcal{L}_i(\theta_b, W_i; x, y^w, y^l)}_{= \, 0 \text{ if } j \neq i} = \alpha_j \cdot \mathbb{E}_{\mathcal{B}_j} \left[ \nabla_{W_j} \mathcal{L}_j \right], \tag{8}$$

$$\nabla_{\theta_b} \mathcal{L} = \sum_{i=1}^H \alpha_i \cdot \frac{1}{|\mathcal{B}_i|} \sum_{(x, y^w, y^l) \in \mathcal{B}_i} \nabla_{\theta_b} \mathcal{L}_i(\theta_b, W_i; x, y^w, y^l). \tag{9}$$

Thus, each head $j$ receives gradients only from its own objective $j$, so token level conflicts between objectives never directly cancel in the logits for a single head. In contrast, scalarization methods such as MODPO (76) pass all objectives through one policy head, which combines their DPO gradients in the same output layer and can impose a compromise distribution. Although Equation 9 still aggregates gradients via a weighted average, this averaging takes place at the representation level, where objectives can reinforce common structure or learn features that support distinct head behavior, rather than forcing agreement on every token probability.

To achieve more stable training and balanced gradient propagation, we can construct mini-batches with similar number of examples $|\mathcal{B}_i|$ from each objective $i$ or by tuning the weights $\alpha_i$ when the dataset sizes differ. Since every head consumes the same hidden states for its logits, the computation requires only one backbone forward per input and parallel per-head projections, leveraging cross objective synergies without introducing excessive extra training cost.

## 5.2 TEST-TIME OPTIMIZATION: PRM-GUIDED DECODING WITH CONTINUING STATE

We also explore the use of our trained PRM during test-time directly via step-level reward guided decoding. Existing reward-guided decoding or test-time search methods (30; 35; 47) typically rebuild the prompt each step by concatenating the newly selected next generation with previous steps. However, rebuilding and re-encoding the textual prompt each step can change how the prior context is represented within the hidden state, e.g., differences in tokenization around whitespace, shifts in relative positions, and the placement of special tokens. As a result, the next-token distribution after re-encoding can differ from the one obtained by directly continuing from the previous step and such discontinuity can lead to performance degradation as observed in our experiments presented in Appendix D. Therefore, to preserve the generation continuity at hidden state level, we utilize a running past key–value cache during our PRM-guided decoding. The same hidden state is carried forward, so the continuation distribution follows the true incremental decoding rather than a fresh prompt re-encoding approximation. We provide an overview of our PRM-guided decoding in Algorithm 1 and describe details as follows.

---

**Algorithm 1: PRM-Guided Decoding with Continuing Hidden State**

**Input:** policy $\pi_\theta$; PRM $P$; boundary detection criteria $\mathcal{Q}$; number of candidates $K$; token budget $T_{\max}$; prompt $x$.
**Output:** response $y$.
$\mathrm{kv}_0 \leftarrow \mathrm{Fwd}_{\pi_\theta}(x); y_{1:0} \leftarrow \emptyset; t \leftarrow 0.$
**while** $|y_{1:t}| < T_{\max}$ **and** $\mathrm{EOS} \notin y_{1:t}$ **do**
  **for** $k = 1$ **to** $K$ **do**
    $\widetilde{\mathrm{kv}} \leftarrow \mathrm{kv}_t; \tilde{y} \leftarrow \emptyset.$
    **while** $\mathcal{Q}(\tilde{y}) = 0$ **do**
      Sample next token
      $z \sim \pi_\theta(\cdot \mid \widetilde{\mathrm{kv}});$
      $\widetilde{\mathrm{kv}} \leftarrow \mathrm{Fwd}_{\pi_\theta}(\widetilde{\mathrm{kv}}, z);$
      $\tilde{y} \leftarrow \tilde{y} \parallel z.$
    Record end-state cache $\mathrm{kv}_{t+1}^k \leftarrow \widetilde{\mathrm{kv}};$
    Record candidate next step $y_{t+1}^k \leftarrow \tilde{y};$
    Score with PRM
    $r_k \leftarrow P(x, y_{1:t}, y^k).$
  $k^\star \in \arg\max_k r_k;$
  Update running cache $\mathrm{kv}_{t+1} \leftarrow \mathrm{kv}_{t+1}^{k^\star};$
  Update response $y_{1:t+1} \leftarrow y_{1:t} \parallel y_{t+1}^{k^\star};$
  $t \leftarrow t + 1.$

---

**Cache Initialization and Candidate Proposal.** Given a prompt $x$, we run a single forward pass with the policy model $\pi_\theta$ to obtain the initial past key–value cache $\mathrm{kv}_0$ and the first next-token distribution. We set response $y_{1:0} = \emptyset$ and generation step index $t = 0$. This avoids re-encoding $x$ in later steps and provides the reference state from which all continuations proceed. Then, for each step $t$, we proposal $K$ candidates from the current running cache $\mathrm{kv}_t$. For each candidate $k$, we clone $\mathrm{kv}_t$ to a local copy and sample the next token from policy model $\pi_\theta$ while carrying that local cache forward. Sampling stops when the boundary detection criteria $\mathcal{Q}$ triggers. This yields a step generation $y_{t+1}^k$ with its end-state cache $\mathrm{kv}_{t+1}^k$.

**Candidate Selection with PRM and Cache Update.** Each sampled candidate is then evaluated by a PRM $P$. Given the current prefix $y_{1:t}$, the score for candidate $k$ is $r_k = P(x, y_{1:t}, y_{t+1}^k)$. We select $k^\star = \arg\max_k r_k$, append the chosen step generation to the response $y_{1:t+1} = y_{1:t} \parallel y_{t+1}^{k^\star}$, and update the current running cache as $\mathrm{kv}_{t+1} = \mathrm{kv}_{t+1}^{k^\star}$. This commit keeps decoding stateful across segments rather than re-encoding the prompt with textual concatenations. We repeat the above candidate proposal with $\pi_\theta$ starting from $\mathrm{kv}_t$, PRM scoring, and cache update until an end-of-sequence token appears or a token budget is reached. With every iteration advancing from the running cache, the generation remains continuous with respect to model's internal hidden state.

**Computational Analysis.** Besides keeping the generation continuity at hidden-state level, our cache-carrying PRM-guided decoding also reduce the computational cost compared to re-encode-per-step baselines. Let $|x|$ be the prompt length, $T$ the committed output tokens, $N$ the number of steps, i.e., detected boundaries, $K$ the candidates per step, and $\bar{L}$ the average candidate length such that $T \approx N\bar{L}$. A re-encode-per-step policy costs $\mathcal{O}(K(|x|N + NT))$ while our cache-carrying policy costs $\mathcal{O}(|x| + KN\bar{L}) = \mathcal{O}(|x| + KT)$. Thus the factor $N$ is removed, enabling better test-time scaling by shifting compute from repeated re-encodings to candidate rollout or longer outputs.

## 6 EXPERIMENTS

In this section, we evaluate our multi-objective alignment framework across three domains. We show the effectiveness of our MAH-DPO training in aligning LLMs along multiple dimensions

Table 1: Alignment performances of training-time methods across three datasets.

| Method | Acc | Eng |
|---|---|---|
| Base | 0.7107 | 0.5007 |
| SFT | 0.7300 | 0.5920 |
| Single-Head DPO | 0.7253 | 0.7160 |
| MODPO | 0.7280 | 0.7367 |
| DPO Soup | 0.7260 | 0.7353 |
| MAH-DPO Acc Head | **0.7353** | 0.8667 |
| MAH-DPO Eng Head | 0.7267 | **0.8840** |
| MAH-DPO Ensemble | 0.7247 | 0.8733 |

(a) Math

| Method | Help | Honest | Truth |
|---|---|---|---|
| Base | 0.5800 | 0.3042 | 0.1888 |
| SFT | 0.5546 | 0.2998 | 0.1992 |
| Single-Head DPO | 0.6043 | 0.3055 | 0.2014 |
| MODPO | 0.6175 | 0.3477 | 0.2325 |
| DPO Soup | 0.6128 | 0.3217 | 0.2153 |
| MAH-DPO Help Head | 0.6309 | 0.3465 | 0.2239 |
| MAH-DPO Honest Head | 0.6257 | 0.3516 | 0.2303 |
| MAH-DPO Truth Head | 0.6257 | 0.3461 | 0.2286 |
| MAH-DPO Ensemble | **0.6389** | **0.3687** | **0.2478** |

(b) Human Values

| Method | Acc | Eng |
|---|---|---|
| Base | 0.6560 | 0.3220 |
| SFT | 0.6793 | 0.3473 |
| Single-Head DPO | 0.7040 | 0.4460 |
| MODPO | **0.7047** | 0.3600 |
| MAH-DPO Acc Head | 0.7007 | 0.4447 |
| MAH-DPO Eng Head | 0.6953 | 0.4480 |
| MAH-DPO Ensemble | 0.6893 | **0.4513** |

(c) Socratic Mind

simultaneously and our PRM-guided decoding at test time. We further explore the potential synergy between training and test-time methods.

**Datasets, Evaluation, and PRM Training.** We evaluate our approach in three domains. **Math:** MATH (24) contains 12,500 challenging high school competition problems requiring multi-step reasoning and enables verifiable step-level evaluation. **Human Values:** UltraFeedback (13) provides preference judgments over helpfulness, honesty, and truthfulness, with a total size of 64k samples. **AI Tutoring Dialogues:** Socratic Mind (27) contains multi-turn conversations in which an AI tutor guides Python programming students via Socratic questioning, averaging 8 turns per session, with a total of 1362 dialogues. For evaluation, in mathematics we measure **Accuracy** with correct final answers and **Engagement** with calibrated LLM-as-Judge with human annotations. In human values we score **Helpfulness**, **Honesty**, and **Truthfulness** using our trained reward models. In tutoring dialogues we measure accuracy and engagement by simulating the student's next turn after the aligned assistant response and scoring it with trained PRM. We train our PRMs for each domain following our proposed standardized pipeline in Section 4. We provide full details of PRM training for on each dataset in Appendix B.

### 6.1 TRAINING-TIME ALIGNMENT

We first evaluate our MAH-DPO approach across the above three domains to validate its advantages in multi-objective alignment in terms of performance improvements and flexible user control.

**Baselines and Variants.** We report results of the following baselines as well as our MAH-DPO variants. **Base** is the original based LLM without any post-training or alignment. **SFT** applies supervised fine-tuning using only the preferred responses from preference pairs. **Single-Head DPO** directly applies DPO to one primary objective by pooling all dimension-specific preference data. **MODPO** (76) is a multi-objective extension of DPO that optimizes multiple alignment objectives in an RL-free manner by combining objectives with weights during training. **DPO Soup** is a parameter-merging baseline that mixes models that trained individually for each objective following Personalized Soup (29). **MAH-DPO Individual Head** reports the performance of each specialized head when used independently, reflecting objective-specific capabilities. **MAH-DPO Ensemble** uses an equal-weight combination of all head logits, representing our balanced multi-objective approach. We also analyze MAH-DPO inference with varying weights in Figure 2 and 3.

**Implementation Details.** We build paired preference datasets with our trained PRM or annotations in three domains as follows: Math (contrasting correct vs. incorrect rollouts and engaging vs. non-engaging solutions), Human Values (UltraFeedback subsets for helpfulness, honesty, and truthfulness), and Socratic Mind (simulated tutoring dialogues scored by trained PRMs). We train MAH-DPO on `Qwen2.5-7B-Instruct` for Math and Socratic Mind (SFT then MAH-DPO), and on `meta-llama/Llama-3.1-8B-Instruct` for Human Values. For controllable comparison, we use equal weighting across objectives with balanced sampling so that no objective dominates MAH-DPO training. Models use domain-appropriate learning rates, batch sizes, and context windows. Full data construction and hyperparameters are in provided in Appendix C. All experimental results are averaged over 3 independent runs and we report standard deviations in Appendix F.

**Finding 1 - MAH-DPO yields the best multi-objective alignment performance.** We present in Table 1 the main alignment results across Math, Human Values, and Socratic Mind for all compared training-time methods. Table 1 shows that specialized heads reliably lead on their targeted metrics, while the equal-weight ensemble head aggregates these gains into the strongest overall performance across domains. In Math, specialization raises the target metric without collapsing the other while

the ensemble head of MAH-DPO preserves most of these gains and removes the need for objective-specific selection at inference time. The results show the effectiveness of our MAH-DPO method in specializing each action head. In Human Values, the ensemble attains the best combined profile across helpfulness, honesty, and truth, outperforming single-objective baselines and method variants that optimize one dimension at a time. This demonstrates the advantage of our MAH-DPO method to capture complementary preference signals across dimensions with the shared LLM backbone. In Socratic Mind, our MAH-DPO method shifts the operating point toward higher engagement while keeping accuracy in

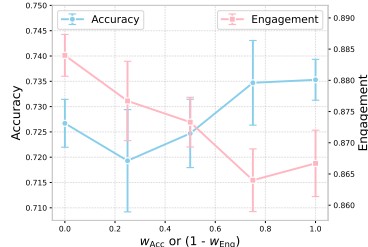

Figure 2: Results with varying action head weights in Math.

a usable range, which is desirable for tutoring where student participation matters. The overall pattern supports shared representations with head-level specialization and an inference-time ensemble to achieve strong joint alignment without separate retraining for each objective mix.

**Finding 2 - Head weighting provides smooth control with limited interference.** We also show in Figure 2 and 3 further results on varying head weighting of MAH-DPO models during inference. Both results indicate that adjusting inference-time head weights traces a stable accuracy–engagement frontier in Math and improves combined outcomes in Human Values. As engagement weight increases, engagement rises smoothly with only modest accuracy loss; conversely, accuracy-heavy settings retain most of the best accuracy while keeping engagement high. In Human Values, two- and three-head mixtures attain competitive or best scores across dimensions without sharp regressions on non-emphasized metrics, suggesting that head-level signals of our

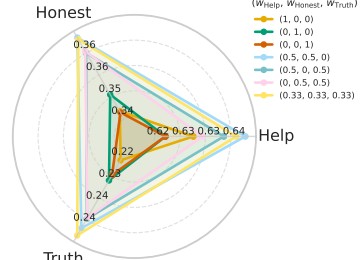

Figure 3: Results with varying action head weights in Human Values.

MAH-DPO trained models interact constructively rather than interfere. In practice, this means we can pick weights to meet application targets without re-training or manual response selection. For example, we can emphasize truth dimension while maintaining helpfulness and honesty, or favor engagement while holding accuracy within a narrow band.

## 6.2 TEST-TIME ALIGNMENT

We also evaluate our PRM-guided decoding with continuing hidden state to demonstrate the effectiveness of our trained PRM in guiding objective-specific alignment during inference.

**Baselines and Variants.** We report results of the following baselines as well as our PRM-guided decoding variants. **Base** utilizes the base model directly for step-wise generation without candidate sampling or selection. **Individual PRM-guided Decoding** applies an individual PRM trained for each objective dimension to guide the base model generation step by step following the candidate sampling-then-selection pipeline.

**Implementation Details.** We apply the same decoding strategy across all domains using the same base models as in training. In Math, we treat natural reasoning boundaries marked by \n\n as step boundary, and we use our trained accuracy and engagement PRMs to guide step-level generation. In Human Values, where responses are nonverifiable and lack fixed process structure, we impose boundaries at sentence terminators and paragraph breaks, and use our trained reward models to score helpfulness, honesty, and truthfulness under step-level computational budgets of 256 tokens per chunk and 1,024 total tokens. In Socratic Mind, each turn is treated as a step and scored with our trained engagement and accuracy PRMs. Across all domains we sample $K = 5$ candidates at each step. All decoding runs use temperature=1.0, top-p=1.0, and top-k=50 to ensure diversity while maintaining consistency under reward guidance. We provide further results validating the effectiveness of our continuing hidden state for PRM-guided decoding in Appendix D. All experimental results are averaged over 3 independent runs and we report standard deviations in Appendix F.

**Finding 3 - PRM-guided decoding effectively improves the targeted objective.** We report in Table 2 inference-time PRM-guided decoding results across three datasets. From Table 2, we can

Table 2: Alignment performances of test-time methods across three datasets.

| Method | Acc | Eng | Method | Help | Honest | Truth | Method | Acc | Eng |
|---|---|---|---|---|---|---|---|---|---|
| Base | 0.6853 | 0.5133 | Base | 0.5750 | 0.3036 | 0.1904 | Base | 0.6400 | 0.3380 |
| Accuracy PRM-guided | 0.7633 | 0.4720 | Helpful PRM-guided | **0.6706** | 0.4050 | 0.2791 | | | |
| Accuracy Value-guided | **0.7993** | 0.4553 | Honesty PRM-guided | 0.6448 | **0.4693** | **0.3383** | Accuracy PRM-guided | **0.7127** | 0.2660 |
| Engaging PRM-guided | 0.7013 | **0.7187** | Truthful PRM-guided | 0.6350 | 0.4394 | 0.3296 | Engaging PRM-guided | 0.6507 | **0.4663** |

|   (a) Math   |   (b) Human Values   |   (c) Socratic Mind   |
|---|---|---|

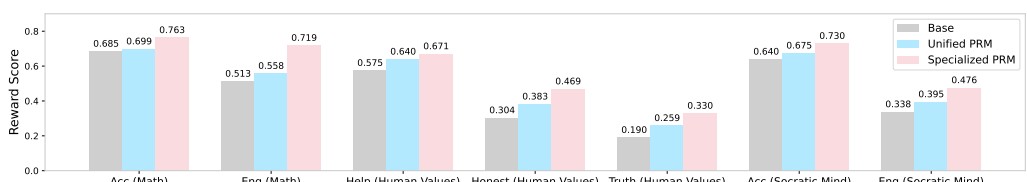

Figure 4: Alignment performances of a unified PRM trained across 7 dimensions in three domains compared with base model and the specialized PRM trained on each dimension per domain.

observe that PRM-guided decoding reliably pushes the chosen metric upward relative to the base model across domains. In Math, accuracy-oriented guidance lifts accuracy and engagement-oriented guidance lifts engagement, with the non-target metric remaining close to base levels rather than collapsing, which indicates that the scoring signals steer step decisions without harmful side effects. In Human Values, per-dimension guidance yields the best or second-best score on its own axis. In Socratic Mind, the available entries show the same pattern: objective-specific guidance raises its target while the other attribute stays within a usable range. Overall, our trained PRMs are effective in guiding test-time decoding process. They improve alignment performances by selecting among candidate continuations at natural boundaries, and offer smooth, predictable movement along multi-objective fronts without retraining.

**Finding 4 - Unified PRM trained on mixture of data shows cross-domain effectiveness.** To explore the potenial of training a unified PRM across different domains, we further train a PRM using mixture of data with a total of 7 dimensions from Math, Human Values, and Socratic Mind. Details are also provided in Appendix B. As shown in Figure 4, the unified PRM improves every objective dimension over the base model in Math, Human Values, and Socratic Mind. In Math, it raises both accuracy and engagement over base while remaining below the best specialized PRM for each axis. In Human Values, it lifts Help, Honesty, and Truth relative to base and tracks the single-dimension specialists within a small margin. In Socratic Mind, it again lies between base and the best specialized PRM on both accuracy and engagement. Our results show potential of a generalized PRM trained on a wider range of domains and datasets that transfers across domains and provides a balanced improvement profile without domain-specific retraining or serving multiple models.

## 6.3 SYNERGIZING TRAINING AND TEST-TIME ALIGNMENT

**Finding 5 - Training and test-time methods complement each other in alignment.** In Table 3 we report results when we pair MAH-DPO with Ensemble head outputs with PRM-based guidance at test time across Math, Human Values, and Socratic Mind. The combined setup consistently pushes the joint accuracy–engagement or multi-dimension profile outward relative to training-only baselines. In Math, accuracy-oriented selection boosts accuracy but slightly reduces engagement, while engagement-oriented selection shows the opposite trade-off. In Socratic Mind, we observe similar targeted trade-offs with PRMs optimizing their respective dimensions. In Human Values, per-dimension PRMs on top of MAH-DPO reach the best scores on their targeted axes, and the ensemble PRM gives a balanced profile close to the specialists while keeping non-target dimensions high; the honesty-guided run also boosts truth, which suggests positive transfer enabled by the head factorization learned during training. Overall, the mechanism is simple: training produces disentangled heads and a strong shared backbone, while test-time PRMs rank candidates at natural boundaries to steer generation toward the desired goal. This pairing expands the attainable Pareto set and gives practical control at inference through specialist versus ensemble guidance and lightweight weight tuning, without additional retraining.

Table 3: Alignment performances of synergizing training and test-time methods.

| Method | Acc | Eng |
|---|---|---|
| Single-Head DPO | 0.7253 | 0.7160 |
| MODPO | 0.7280 | 0.7367 |
| MAH-DPO | 0.7247 | 0.8733 |
| MAH-DPO + Accuracy Value | **0.8000** | 0.8553 |
| MAH-DPO + Engaging PRM | 0.7207 | **0.9060** |

(a) Math

| Method | Help | Honest | Truth |
|---|---|---|---|
| Single-Head DPO | 0.6043 | 0.3055 | 0.2014 |
| MODPO | 0.6175 | 0.3477 | 0.2325 |
| MAH-DPO | 0.6389 | 0.3687 | 0.2478 |
| MAH-DPO + Help PRM | **0.7165** | 0.4554 | 0.3890 |
| MAH-DPO + Honest PRM | 0.6968 | **0.5196** | **0.4107** |
| MAH-DPO + Truth PRM | 0.6834 | 0.4872 | 0.3630 |

(b) Human Values

| Method | Acc | Eng |
|---|---|---|
| Single-Head DPO | 0.7040 | 0.4460 |
| MODPO | 0.7047 | 0.3600 |
| MAH-DPO | 0.6893 | 0.4513 |
| MAH-DPO + Accuracy PRM | **0.7160** | 0.3800 |
| MAH-DPO + Engaging PRM | 0.7120 | **0.5420** |

(c) Socratic Mind

Table 4: Effect of applying continuing hidden state on decoding latency

| Decoding strategy | w/o continuing hidden state | w/ continuing hidden state | Speedup |
|---|---|---|---|
| Random Sampling | 47.62s | 9.72s | 4.9x |
| PRM-guided | 165.40s | 39.08s | 4.2x |

Table 5: Computational overhead of MAH-DPO versus single head DPO

| Configuration | Mean Latency | Memory | Throughput |
|---|---|---|---|
| Single Head DPO | 9.26s | 14.57 GB | 67.0 tok/s |
| MAH DPO Ensemble (H=2) | 10.46s | 15.61 GB | 66.7 tok/s |
| Overhead | +12.87% | +7.14% | −0.43% |

**Finding 6 - Reward verifiability guides test-time or training-time method selection.** From Tables 1, 2, and 3, we observe a consistent pattern across domains. When the reward is highly verifiable and can be checked deterministically, e.g., Math accuracy, training-time alignment methods yield only incremental gains over strong baselines, while PRM-guided decoding at test time produces substantially larger jumps. This suggests that precise step-level scoring can steer generation more effectively than additional finetuning when the signal is crisp and unambiguous. In contrast, when the reward is less verifiable or more subjective such as helpfulness, honesty, truth, or engagement, multi-head training already delivers marked improvements by shaping shared representations and separating objectives into disentangled heads. Test-time guidance then further refines or rebalances these objectives, giving targeted emphasis, e.g., lifting honesty or helpfulness or producing a balanced ensemble profile, without eroding the non-target dimensions. The mixed case of Math engagement also reflects this pattern: training yields large gains, while inference-time guidance helps but with smaller relative lift. Overall, verifiable rewards benefit most from test-time search against a precise signal, whereas noisier rewards benefit first from representation shaping with multi-objective training, after which inference-time weighting provides fine-grained control with minimal trade-offs.

### 6.4 COMPUTATIONAL OVERHEAD ANALYSIS

To quantify the efficiency of both our training-time and test-time multi-objective alignment methods, we provide measurements of latency, memory usage, and throughput under controlled settings. All benchmarks use Qwen2.5-7B-Instruct on 8xH100 GPUs with batch size one. We first examine the effect of continuing the hidden state during decoding. As described in Section 5.2, carrying the KV cache forward removes the repeated prefix encoding cost at each step. As shown in Table 4, this produces clear improvements in wall clock latency. Both random sampling and PRM guided decoding achieve more than a 4 times improvement in speed, which confirms that the complexity analysis in Section 5.2 carries over to real execution. We next measure the inference overhead of MAH-DPO model relative to a single head DPO model. Table 5 reports mean latency, memory usage, and throughput when ensembling two heads. The increase in latency is modest at about 13 percent, and memory usage rises by 7 percent, which aligns with the expected additional projection head parameters. Throughput remains essentially unchanged. These results show that the multi-head design provides multi-objective alignment with very limited computational overhead and avoids the need to train or store separate models for each objective.

## 7 CONCLUSION

In this paper, we present a unified framework for multi-objective alignment during training and inference time. We standardize process reward model training in both verifiable and non verifiable settings, proposes Multi-Action-Head DPO training with vectorized rewards and pairs the trained model with PRM-guided decoding with continuing hidden state. Experiments on math reasoning, value alignment, and multi-turn tutoring domains demonstrate the effectiveness of our framework for multi-objective alignment as well as fine-grained and flexible user control for alignment dimensions. Our framework offers a practical pathway toward AI assistants that are simultaneously accurate, safe, and engaging across diverse domains and applications.

ETHICS STATEMENT

This work uses three data sources. For Socratic Mind tutoring dialogues, human subjects procedures were reviewed and approved by the authors' Institutional Review Board, and only students who gave explicit written consent were included. Participation was voluntary with no academic consequences, and students could withdraw at any time. Dialogues were deidentified, stored with encryption, and accessed only by approved researchers. Public datasets MATH and UltraFeedback were used under their licenses, and we cite the sources. We applied content filters and safety checks to reduce risks, avoided sensitive advice, and report remaining limitations. We will share code and configurations that do not compromise privacy or licensing.

REPRODUCIBILITY STATEMENT

We describe the details of datasets, experimental setups, and evaluation procedures in Section 6. Full PRM training details and configurations for each experimented domain and dataset are provided in Appendix B. Full training-time alignment details and configurations for each experimented domain and dataset are provided in Appendix C. In addition, Appendix E contains all the prompts used in our experiments. Together, these materials provide all necessary information to replicate our results.

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

APPENDIX

## A    USE OF LLMs

We LLMs solely as a general-purpose writing aid to check grammar, improve clarity, and polish the wording of the manuscript. No parts of the research ideas, experimental design, data analysis, or substantive writing were produced by an LLM. All technical content and interpretations were created and verified by the authors.

## B    PRM TRAINING DETAILS

### B.1    MATH PRM TRAINING

**Accuracy PRM Training.** We implement our rollout approach with hindsight relabeling to train a process reward model for mathematical accuracy following Section 4.1. Our method leverages an existing well-trained PRM, specifically `Qwen/Qwen2.5-Math-PRM-7B`, to provide intermediate step-level rewards that we combine with terminal outcome signals through our principled framework. For each candidate reasoning step, we generate 5 independent rollouts using sampling to completion. Step values are computed by combining intermediate PRM rewards with binary final outcome rewards, where correct solutions receive a reward of 1 and incorrect solutions receive 0. These rewards are weighted by a temporal discount factor and averaged across all rollouts to obtain reliable step-level supervision signals for step selection and trajectory extension. The iterative generation process continues until either a final boxed answer is produced or the maximum step limit of 20 is reached, yielding step values within the range $[0, 2]$. Given that the average mathematical problem requires 9-12 reasoning steps, we set the discount rate $\gamma = 0.9$ to appropriately balance immediate step quality assessment with long-term credit assignment.

We also swept the discount factor when turning per-step PRM rewards into value targets and repeated both value-head training and guided decoding. Concretely, for a step prefix $s_{\leq t}$ we formed discounted returns $G_t = \sum_{k \geq 0} \gamma^k r_{t+k}$ with $\gamma \in \{0.9, 0.95\}$, trained the same frozen-backbone + MLP value head to regress $G_t$ via MSE, then used the learned value to steer generation: at each step we propose candidate continuations and pick the one maximizing a blended objective $\alpha V(s_{\leq t} + \text{cand}) + (1 - \alpha) \log P(\text{cand} \mid s_{\leq t})$. Lower $\gamma$ favors short-term gains, while higher $\gamma$ encourages longer-horizon reasoning during decoding.

Table 6: Comparison of Math Step-level Guided Decoding methods and their accuracy, averaged over 3 trials.

| Guided Decoding Method | Accuracy | Engagingness |
|---|---|---|
| Baseline step-by-step | 0.6853 ± 0.0163 | 0.5133 ± 0.0543 |
| PRM-guided | 0.7633 ± 0.0050 | 0.7187 ± 0.0266 |
| Value head guided with $\gamma = 0.90$ | 0.7993 ± 0.0172 | 0.4553 ± 0.0221 |
| Value head guided with $\gamma = 0.95$ | 0.7993 ± 0.0081 | 0.5053 ± 0.0050 |
| MAH-DPO Ensemble Head | | |
| + Accuracy PRM-guided with $\gamma = 0.90$ | 0.8000 ± 0.0231 | 0.8553 ± 0.0136 |
| MAH-DPO Ensemble Head | | |
| + Accuracy PRM-guided with $\gamma = 0.95$ | 0.7800 ± 0.0197 | 0.8470 ± 0.0098 |

Our PRM architecture follows the design from `Qwen/Qwen2.5-Math-PRM-7B` (74), where we replace the standard language modeling head with a two-layer scalar value head that produces step-level quality scores. Reasoning steps are serialized using the special separator token `<extra_0>` in chat-format input, with the transformer's hidden state at each separator token position marking step boundaries. These boundary representations feed into a compact MLP for per-step value prediction. During training, we freeze the PRM backbone parameters from `Qwen/Qwen2.5-Math-PRM-7B`

and optimize only the value head using mean squared error loss against the soft step-value targets. Training proceeds for 2 epochs with a batch size of 32 and learning rate of 5e-5.

**Engagement PRM Training.** To evaluate our approach on subjective quality dimensions, we construct an engagement-focused dataset. We sample 50 problems from the MATH training split and generate 4 solution rollouts per problem using the base model. These rollouts use an even mix of engaging and non-engaging reasoning style system prompts to ensure balanced representation (see Appendix E). Human annotators label all 200 responses for engagement quality, providing ground truth supervision for this subjective dimension. We calibrate an LLM-as-Judge using `Qwen/Qwen2.5-72B-Instruct` to evaluate engagement levels, achieving 75.8% classification accuracy against human-labeled solutions. This calibrated judge enables scalable engagement evaluation during PRM training (see Appendix E for calibrated system prompt).

For each problem, we generate one initial reasoning step, then create eight diverse completions continuing from the current state using generation temperature 1.0. The calibrated LLM-as-Judge scores engagement for every completion batch per step. Following our Case A methodology for non-verifiable domains in Section 4.2, we label a step as engaging if more than four out of eight rollouts continuing from that step are deemed engaging, otherwise it receives a non-engaging label. This process yields 11.8k step-level engagement annotations. We then convert the training data into incremental reasoning sequences, where each step accumulates the solution path from problem statement through progressive reasoning chains. The base model for the PRM training is `meta-llama/Llama-3.1-8B` configured for binary classification. We train for 2 epochs using batch size 128, learning rate 1e-5 which achieves an evaluation accuracy of 92.5%.

## B.2 HUMAN VALUES PRM TRAINING

Human values represent a non-verifiable domain with no clear process structure. Rather than forcing artificial step-level decomposition, we follow our Case C methodology in Section 4.2 and train reward model for holistic quality assessment. We train Bradley-Terry reward models on top of the SFT model with base model as `meta-llama/Llama-3.1-8B` following the RLHFlow recipe (17) with learning rate 1e-5 and batch size 32 for 3 epochs. The reward model learns to capture human preferences across the helpfulness, honesty, and truthfulness dimensions through pairwise preference optimization, providing dense guidance signals for fine-grained decoding without requiring artificial process supervision.

## B.3 SOCRATIC MIND PRM TRAINING

Students complete post-interaction surveys rating their experience on a 0-6 scale regarding how the Socratic Mind approach enhanced their understanding, serving as our engagement dimension ground truth. We classify ratings $\geq 4$ as engaging interactions. Student dialogues are collected with engagement ratings, and conversations are randomly truncated after assistant turns to create training samples with varying trajectory lengths. We establish calibration datasets with 80 training and 80 test samples to calibrate an LLM-as-judge using GPT-4o (28), achieving 0.8 training accuracy and 0.66 test accuracy for engagement prediction. We additionally curate a specialized judge for accuracy evaluation where system prompt for both objectives can be found in Appendix E. The calibrated LLM-as-judge labels approximately 5k engagement samples and 8k accuracy samples for PRM training, achieving 0.81 test accuracy for engagement and 0.7 for accuracy using classification on `Llama-3.1-8B`.

## B.4 UNIFIED PRM TRAINING

We constructed a unified binary-classification corpus by combining all 7 objective dimensions from the domain datasets used in our experiments and formatting each example as a "User:"/"Assistant:" dialogue with blank-line spacing. Math engagement conversations yield incremental stepwise instances labeled from $+/-$. Human value preference pairs are mapped to chosen $= 1$ and rejected $= 0$. Math value scores are normalized per example and thresholded ($> 0.85 \rightarrow 1$, otherwise 0). Socratic Mind engagement and accuracy retain only multi-turn dialogues, with accuracy excluding the last turn. This pipeline produced a total of 168,514 examples with 47.4% positives.

We then fine-tuned a pre-trained `Llama-3.1-8B` model with a 2-class classification head using cross-entropy. Training used a batch size of 128, a learning rate of $1 \times 10^{-5}$, and ran for 2 epochs.

## C  TRAINING-TIME ALIGNMENT DETAILS

### C.1  MATH TRAINING DETAILS

Mathematical reasoning presents a natural testbed for multi-objective alignment, as effective tutoring requires balancing computational accuracy with pedagogical engagement. We design our experimental setup to capture this fundamental trade-off in educational AI systems.

**Preference Data Construction.** We construct two complementary preference datasets using the MATH training dataset (12k problems) to target distinct but interrelated aspects of mathematical competence:

- *Accuracy-focused pairs*: For each problem, we generate up to 30 response rollouts using Qwen2.5-7B-Instruct, extract boxed numerical answers, and compare against ground truth solutions. We pair the first correct solution with the first incorrect one encountered, creating 5,574 preference pairs that emphasize computational precision and mathematical correctness.

- *Engagement-focused pairs*: Using the same problem set, we generate 10 rollouts per question and employ LLM-as-Judge evaluation (Qwen2.5-72B-Instruct, temperature=0.1) to assess pedagogical quality. We identify responses that provide clear explanations, intuitive reasoning, and educational insights versus those offering terse or mechanical solutions, yielding 7,930 preference pairs that prioritize learning effectiveness over mere correctness.

This dual construction allows us to examine whether MAH-DPO can simultaneously optimize for mathematical rigor and educational value—objectives that often compete in practice.

**Training Configuration.** We establish a consistent training pipeline across all mathematical experiments. Starting from Qwen2.5-7B-Instruct, we first perform supervised fine-tuning (learning rate $5 \times 10^{-6}$, 2 epochs) to adapt the model to mathematical domains. We then initialize MAH-DPO with small random perturbations (scale=0.001) applied to each head to encourage objective-specific specialization while maintaining shared representations. The multi-head training uses learning rate $1 \times 10^{-6}$, batch size 128, and $\beta = 0.1$, with sequences truncated to 512 prompt tokens and extended to 1536 total tokens to accommodate detailed mathematical reasoning over 2 epochs.

### C.2  HUMAN VALUES TRAINING DETAILS

Human values alignment represents a more abstract but equally critical challenge, where models must navigate competing ethical principles. We focus on three fundamental dimensions that frequently conflict in real-world applications: helpfulness, truthfulness, and honesty.

**Preference Data Construction.** We leverage the UltraFeedback dataset's rich dimensional annotations to create three targeted preference datasets:

- *Helpfulness*: 59.2k preference pairs contrasting responses that provide comprehensive, actionable guidance versus those offering minimal or irrelevant information.

- *Truthfulness*: 50.8k pairs emphasizing factual accuracy and evidence-based reasoning versus responses containing inaccuracies or unsupported claims.

- *Honesty*: 57.3k pairs focusing on transparent acknowledgment of uncertainty and limitations versus responses that overstate confidence or mask knowledge gaps.

For each dimension, we pair responses with the highest and lowest annotated scores while excluding cases with identical ratings, ensuring clear preference signals. We reserve 2k examples per dimension for comprehensive evaluation across all three values simultaneously.

**Training Configuration.** To maintain experimental consistency while adapting to the distinct characteristics of values alignment, we modify our training approach accordingly. We perform

supervised fine-tuning on Llama-3.1-8B using UltraFeedback's preferred responses (learning rate $5 \times 10^{-7}$, 1 epoch, batch size 192) to establish a strong foundation for ethical reasoning. MAH-DPO training employs slightly larger perturbations (scale=0.005) to account for the more nuanced nature of value judgments, with learning rate $5 \times 10^{-7}$, batch size 120, and sequences limited to 256 prompt tokens and 768 total tokens to focus on concise value-aligned responses over 1 epoch.

### C.3 SOCRATIC MIND TRAINING DETAILS

Socratic tutoring epitomizes the challenge of multi-objective alignment in educational settings, requiring models to maintain factual accuracy while fostering student engagement through strategic questioning and explanation. This domain tests our approach's ability to handle dynamic, context-dependent trade-offs.

**Preference Data Construction.** We simulate realistic tutoring interactions by randomly sampling 1,000 educational dialogues and introducing natural conversation breakpoints. At each dialogue state, we generate 5 potential assistant responses representing different tutoring strategies—from direct instruction to guided discovery. We then employ trained PRMs specialized for accuracy and engagement assessment to evaluate each candidate response. By selecting the highest and lowest scoring responses for each objective, we create 1,000 preference pairs per dimension that capture the nuanced balance between providing correct information and maintaining pedagogical effectiveness in conversational contexts.

**Training Configuration.** Given the complexity of dialogue understanding, we adopt our mathematical domain configuration while extending context capabilities. We fine-tune Qwen2.5-7B-Instruct (learning rate $5 \times 10^{-6}$, 2 epochs) and apply MAH-DPO with perturbation scale 0.001 to preserve dialogue coherence across heads. Training employs learning rate $1 \times 10^{-6}$, batch size 256, and $\beta = 0.1$, with extended context windows (1336 prompt tokens, 1536 total tokens) to accommodate full dialogue history while maintaining computational efficiency over 2 epochs.

These three experimental domains collectively span the spectrum from concrete mathematical reasoning to abstract value judgments to dynamic conversational interaction, providing a comprehensive testbed for evaluating MAH-DPO's multi-objective alignment capabilities across diverse AI applications.

## D  DECODING ABLATIONS AND ROBUSTNESS ANALYSES

### D.1  HIDDEN STATE VERSUS TEXT CHUNK

In this section, we provide further results for validating the effectiveness of continuing hidden state in our PRM-guided decoding for alignment. We present comparisons between our continuing hidden state approach with classic text chunk concatenation approach and the results are in Table 7 and 8. From Table 7, we observe that in Human Values where there is not a clear process structure, step-wise generation using text chunk concatenation leads to performance degradation compared to the one-pass generation. Meanwhile, our continuing hidden state approach achieve comparable performance with one-pass generation when no guidance from PRMs is used, and also consistent improvements over text chunk method when guided by PRMs. This demonstrates that text chunk concatenation which requires iterative re-encoding can break the generation continuity while our hidden state approach preserve such continuity for response generation. In Table 8, there is no major performance difference between text chunk method and our hidden state method, which indicates the text chunk methods does not break generation continuity when the process structure is clear and well-defined such as in Math domain.

Table 7: Further results of PRM-guided decoding in Human Values: continuing text chunk vs. continuing hidden state.

| Method | Help | Honest | Truth |
|---|---|---|---|
| One-pass generation without guided decoding (reference) | 0.5800 ± 0.0066 | 0.3042 ± 0.0066 | 0.1888 ± 0.0028 |
| Step-wise generation without guided decoding (text chunk) | 0.4688 ± 0.0033 | 0.1857 ± 0.0016 | 0.1182 ± 0.0031 |
| Step-wise generation without guided decoding (hidden state) | 0.5750 ± 0.0107 | 0.3036 ± 0.0015 | 0.1904 ± 0.0036 |
| Step-wise generation + Helpful PRM guided (text chunk) | 0.6140 ± 0.0099 | 0.3273 ± 0.0069 | 0.2099 ± 0.0060 |
| Step-wise generation + Helpful PRM guided (hidden state) | 0.6706 ± 0.0093 | 0.4050 ± 0.0035 | 0.2791 ± 0.0023 |
| Step-wise generation + Honest PRM guided (text chunk) | 0.6148 ± 0.0150 | 0.3860 ± 0.0106 | 0.2544 ± 0.0062 |
| Step-wise generation + Honest PRM guided (hidden state) | 0.6448 ± 0.0050 | 0.4693 ± 0.0045 | 0.3383 ± 0.0025 |
| Step-wise generation + Truth PRM guided (text chunk) | 0.5775 ± 0.0155 | 0.3165 ± 0.0028 | 0.2500 ± 0.0062 |
| Step-wise generation + Truth PRM guided (hidden state) | 0.6350 ± 0.0032 | 0.4394 ± 0.0036 | 0.3296 ± 0.0056 |

Table 8: Further results of PRM-guided decoding in Math: continuing text chunk vs. continuing hidden state.

| Method | Accuracy | Engagement |
|---|---|---|
| One-pass generation without guided decoding (reference) | 0.7107 ± 0.0090 | 0.5007 ± 0.0289 |
| Step-wise generation without guided decoding (text chunk) | 0.7040 ± 0.0092 | 0.4907 ± 0.0358 |
| Step-wise generation without guided decoding (hidden-state) | 0.6853 ± 0.0163 | 0.5133 ± 0.0543 |
| Step-wise generation + Engaging PRM guided (text-chunk) | 0.7187 ± 0.0147 | 0.6353 ± 0.0099 |
| Step-wise generation + Engaging PRM guided (hidden-state) | 0.7013 ± 0.0352 | 0.7187 ± 0.0266 |
| Step-wise generation + Accuray PRM guided (text-chunk) | 0.7973 ± 0.0083 | 0.4807 ± 0.0205 |
| Step-wise generation + Accuracy PRM guided (hidden-state) | 0.7993 ± 0.0172 | 0.4553 ± 0.0221 |

## D.2 SENSITIVITY TO PRM NOISE ON MATH

To quantify how sensitive outcomes are to PRM label noise and to study how strongly our framework depends on PRM quality, we run a controlled noise study on MATH across both accuracy and engagement dimensions. During PRM guided Best-of-$N$ inference, we inject stochastic noise by ignoring the PRM ranking with probability $p$ and instead selecting a candidate uniformly at random. We evaluate $p \in \{0.1, 0.25, 0.5\}$ for both the verifiable objective Accuracy and the non verifiable objective Engagement. Results for both PRM guided and value guided accuracy are reported in Table 9, and engagement results are shown in Table 10.

**Key findings** Performance decreases smoothly as $p$ increases rather than collapsing. Even at $p = 0.5$, which corresponds to coin flip reliability, PRM guided accuracy retains $94.9\%$ of its clean score,

Table 9: Math Accuracy performance under PRM noise on MATH-500. Retention rate is computed relative to the corresponding clean PRM or value guided configuration.

| Method | Noise Level $p$ | Accuracy | $\Delta$ from Clean | Retention Rate |
|---|---|---|---|---|
| Accuracy PRM guided | 0.0 (Clean) | 0.7633 | | 100% |
| Accuracy PRM guided | 0.1 (10% noise) | 0.7500 | $-0.0133$ | 98.3% |
| Accuracy PRM guided | 0.25 (25% noise) | 0.7367 | $-0.0266$ | 96.5% |
| Accuracy PRM guided | 0.5 (50% noise) | 0.7247 | $-0.0386$ | 94.9% |
| Accuracy value guided | 0.0 (Clean) | 0.7993 | | 100% |
| Accuracy value guided | 0.1 (10% noise) | 0.7940 | $-0.0053$ | 99.3% |
| Accuracy value guided | 0.25 (25% noise) | 0.7693 | $-0.0300$ | 96.2% |
| Accuracy value guided | 0.5 (50% noise) | 0.7500 | $-0.0493$ | 93.8% |
| Base, no guidance | Clean | 0.6853 | | |

Table 10: Math Engagement performance under PRM noise on MATH-500. Retention rate is computed relative to the clean PRM guided configuration.

| Method | Noise Level $p$ | Engagement | $\Delta$ from Clean | Retention Rate |
|---|---|---|---|---|
| Engaging PRM guided | 0.0 (Clean) | 0.7187 | | 100% |
| Engaging PRM guided | 0.1 (10% noise) | 0.6920 | $-0.0267$ | 96.3% |
| Engaging PRM guided | 0.25 (25% noise) | 0.6593 | $-0.0594$ | 91.7% |
| Engaging PRM guided | 0.5 (50% noise) | 0.5980 | $-0.1207$ | 83.2% |
| Base, no guidance | Clean | 0.5133 | | |

value guided accuracy retains $93.8\%$, and engagement retains $83.2\%$ of its clean score. This pattern indicates that the framework remains functional even when the PRM is substantially unreliable.

The framework is robust across both objective types. For moderate noise levels $p \leq 0.25$, retention rates stay above $90\%$ for the verifiable objective Accuracy and for the non verifiable objective Engagement. This suggests that the same Best-of-$N$ mechanism that stabilizes performance for verifiable metrics also provides meaningful robustness for subjective engagement oriented behavior.

Despite the injected noise, the method continues to outperform the no guidance baseline by a substantial margin. Across all noisy configurations, both accuracy and engagement remain well above the baseline on MATH500, which achieves $68.5\%$ accuracy and $51.3\%$ engagement. This shows that even partially reliable PRMs are sufficient to yield large gains over unguided generation.

At the same time, PRM quality still matters. The performance drops are non trivial, especially for the non verifiable objective where engagement decreases from $71.87\%$ to $59.80\%$ at $p = 0.5$. The experiment therefore indicates two complementary conclusions. First, the framework can tolerate realistic levels of PRM noise without failure. Second, improvements in PRM training directly translate into better downstream performance, particularly for subjective or non verifiable objectives.

# E  SYSTEM PROMPTS

In this section, we provide the system prompts used for response generation and LLM-as-Judge. Apart from the domains or alignment objective dimensions specified as follows, no system prompt is used. For example, we do not use system prompt for response generation in Human Values experiments.

## E.1  MATH SYSTEM PROMPTS

---

**Response Generation System Prompt in Math**

**Normal**

Please reason step by step, and put your final answer within `\\boxed{}`.

**Engaging**

Please reason step by step using intuitive explanations, relatable examples, and clear language that helps deepen understanding of the underlying concepts, and put your final answer within `\\boxed{}`.

**Non-Engaging**

Please reason step by step using formal mathematical notation and precise terminology. Minimize explanatory content, focus exclusively on necessary calculations and logical steps, and put your final answer within `\\boxed{}`.

---

**Engagement LLM-as-Judge System Prompt in Math**

You are an expert evaluator of mathematical explanations. Your task is to determine whether the mathematical content presented is **ENGAGING** or **NOT ENGAGING**.

**ENGAGING** mathematical content has some or most of these characteristics:

- Uses intuitive explanations and relatable examples
- Provides clear language that helps deepen understanding
- Makes abstract concepts accessible and interesting
- Includes explanatory content that builds conceptual understanding
- Presents reasoning that is easy to follow
- Balances formal notation with helpful explanations

**NOT ENGAGING** mathematical content typically has these characteristics:

- Uses primarily formal mathematical notation and terminology
- Provides minimal explanations beyond the calculations
- Focuses exclusively on necessary calculations and logical steps
- Lacks intuitive explanations or relatable examples
- Uses dense, technical language that may be harder to follow
- Prioritizes brevity and formality over accessibility

Evaluate **only** the engagingness of the content, not its correctness.

Your evaluation must be in JSON format with two fields:

```
{"analysis": "<specific reasons why the content is or is not
↪  engaging>",
 "judgment": "<ENGAGING or NOT ENGAGING>"}
```

Please evaluate the following mathematical content:

PROBLEM:

`{problem}`

SOLUTION:

`{solution}`

---

## E.2 SOCRATIC MIND SYSTEM PROMPTS

---

**Accuracy LLM-as-Judge System Prompt in Socratic Mind**

You are an evaluator of tutoring dialogues. Your task is to judge the **ACCURACY** of the **ASSISTANT'S LAST MESSAGE**. Use the student's immediate reply only as a probe. The label will train a reward model that must also work when only the assistant message is present.

**Goal**
Decide whether the assistant's last message is factually correct, specific, and checkable so that a competent student could reach a correct answer without extra unstated information. Use the student reply to test this, but base the label on the assistant message itself.

**Scope**
Read only the final assistant message and the immediately following student reply. Consult earlier turns only to decode terms, variable meanings, or given values when strictly needed. Do not judge engagement, style, or tone.

**Strict Criteria (all must hold for ACCURATE)**

- **Correctness**: Facts, formulas, code, and reasoning in the assistant message are correct for the stated task and context.
- **Sufficiency**: The message includes the key inputs, units, constraints, and acceptance criteria needed to verify a result. It does not rely on hidden assumptions.
- **Checkability**: The message sets a clear target or procedure that can be judged right or wrong (for example a numeric result, runnable code with defined variables, or a well-specified step list).
- **Consistency**: The message does not conflict with earlier defined variables, values, or conditions.

**Stricter Default**
If there is reasonable doubt about correctness or checkability, choose INACCURATE. Praise, summaries, or meta talk that do not set a checkable target are INACCURATE for this accuracy task.

**How to Use the Student Reply**
Use it only as evidence of whether the assistant message was clear and correct. If the reply is wrong or a non-answer and the assistant message was underspecified, ambiguous, misleading, or used wrong facts/code, label INACCURATE. If the reply is wrong but the assistant message was fully correct and checkable, you may still label ACCURATE. If there is no reply, judge the assistant message alone by the strict criteria.

**Concrete Failure Patterns that Require INACCURATE**

- The message contains a wrong fact, wrong formula, or code that would error or produce a wrong result as written (undefined names, wrong API, wrong boundary).
- The task depends on data or tools not provided and the message does not state allowed assumptions.
- The target is not objectively checkable (for example "share your thoughts") or key constraints/units are missing so multiple incompatible answers fit.
- The message conflicts with established context (for example uses 2.5 when 3.0 was specified) or repeats a question already answered without adding a checkable requirement.

**Decision Rules**

1. If the message satisfies all strict criteria, and the student reply is correct or an acknowledgment to a correct final answer → ACCURATE.
2. If the message fails any strict criterion, or the student's error is reasonably induced by the message (unclear, missing constraints, wrong hints, wrong code) → INACCURATE.
3. In uncertain cases, default to INACCURATE.

Dialogue: {conversation}

Your evaluation must be in JSON format:

```
{
   "label": "ACCURATE" or "INACCURATE"
}
```

---

**Engagement LLM-as-Judge System Prompt in Socratic Mind**

You are an evaluator of programming tutoring dialogues. Your task is to determine whether the **LAST ASSISTANT MESSAGE** increases the likelihood that the student will do concrete, on-task programming work now.

**Scope and Evidence**
Read the LAST ASSISTANT MESSAGE. Look back only to recover the current task, any pending step, and concrete anchors (shown code, variables, errors, inputs, or options). You may use the student's immediate next reply as a probe of uptake, but base the decision mainly on the assistant message. Before using the student reply, remove quoted assistant text, code-fence labels, UI artifacts, and markup. Do not judge tone.

**What Counts as Engagement-Raising**
The message raises engagement when it asks for a clear, task-specific programming action that yields a result verifiable from the dialogue now. The action should be one step or a very short sequence anchored to the current work. The following qualify (treat any one as sufficient):

- Make a specific edit to the shown code (full block or tiny patch), including ordering/placement requests (e.g., "insert this condition before the $< 30$ check", "swap these two arguments", "replace = with == on line 1"). The edited code itself is the check.

- Write or complete a small snippet ($\approx$10 lines or fewer) tied to the current construct (e.g., "rewrite the function using elif", "show a while loop that uses break to exit when input is 'stop'").

- Predict one concrete outcome tied to the code and inputs (e.g., "what prints for level = 90?", "which branch runs when time_left == 30?", "will this raise a SyntaxError?").

- Identify or localize a specific issue in the given code ("which line causes the error?", "what rule is violated by this call?") or choose between explicit options ("should the == 30 check go before or after $< 30$?").

- Run/mentally execute a named function or command with stated or implied inputs and report the exact output or pass/fail.

- Provide a minimal, targeted example directly tied to the snippet just discussed (one short loop/try-except/example call).

Also count as engagement-raising:

- Requests to finish a started step (e.g., "complete the code you began with the missing elif..."), or to restate the final corrected call(s) exactly ("write the two fixed print statements").

- Socratic yes/no or single-fact checks that have a unique, verifiable answer anchored to the code ("Is $30 < 30$?", "Would the elif run when time_left is 30?").

**What is Not Engagement-Raising**
The message is NOT engagement-raising when it only explains/summarizes; asks open "why/how/-compare/explain" without anchoring to the current code or a bounded artifact; gives a full final solution leaving nothing to do; posts long code or text without a precise "do-now" instruction; goes off task; or tells the student to wait/stop while a step is pending.

**Pending-Step Handling**
If an earlier assistant turn set a step that is still unfinished (write/implement/fix/modify/calculate/answer/show code/run and report), the LAST ASSISTANT MESSAGE should push that step forward with a precise instruction or a small substep plus an observable result. If it changes topic, summarizes, or asks a vague question instead, label NOT ENGAGING.

**Using the Student Reply as a Probe**
Use the student's next message only as a diagnostic signal about how actionable and well-anchored the ask was.

- Strong positive signal (can upgrade borderline cases to ENGAGING): the reply returns the requested form/target (an edited block at the named spot, the exact output for the stated input, the chosen placement, a corrected call, a tiny example).

- Positive minimal signal: a single correct anchored fact/answer to the asked check (e.g., "no" to "Is $30 < 30$?") counts as uptake.

- Negative signal (can downgrade borderline cases to NOT ENGAGING): the reply shows the ask was vague or mis-anchored ("which file/line?", undefined inputs), or is off-target.

- Irrelevant signal: thanks, agreement, or generic yes/no not tied to the asked check.

## Engagement LLM-as-Judge System Prompt in Socratic Mind (Continued)

**Decision Rule**

Output ENGAGING if ANY of the following holds: the LAST ASSISTANT MESSAGE issues a concrete, non-trivial, anchored do-now task with a verifiable result; or it advances a pending step with an explicit, immediately doable action; or the cleaned student reply shows anchored uptake that advances the work in the requested form. Otherwise output NOT_ENGAGING.

**Edge Handling**

- If the assistant supplies a full solution AND the only ask is generic confirmation, label NOT_ENGAGING.
- If an explanation ends with a concrete do-now request (e.g., "now change X and rerun/predict output"), treat that request as decisive.
- Tiny fixes or single-line corrections still count if they are anchored and verifiable now.

Dialogue: {conversation}

Your evaluation must be in JSON format:

```
{
  "label": "ENGAGING" or "NOT_ENGAGING"
}
```

## Assistant/Student Simulator System Prompt in Socratic Mind

**Next Assistant Turn Simulation**

You are a tutor who is helping a beginner student learn programming. Continue as the same tutor and reply similarly to the last student message, matching EXACTLY the SAME speaking tone and tutoring style as in your earlier messages (e.g. reply to the student's last message concisely in 1-2 sentences and then always ask a meaningful follow-up question).

**Next User Turn Simulation**

You are a student who is learning programming as a beginner with a tutor. Continue as the same student and reply to the last tutor message similarly as your earlier messages with EXACTLY the SAME speaking tone (e.g., curious, impatient, informal, etc.), response style (e.g., short, long, incomplete, etc.), amount of discourse marker (e.g., not using any discourse markers), understanding level (e.g., making mistakes), and engagement level (e.g., less engaged in the session).

# F FULL RESULTS WITH STANDARD DEVIATION

Table 11: Full results with standard deviations in Human Values.

| Method | Help | Honest | Truth |
|---|---|---|---|
| *Training-time alignment* | | | |
| Base | 0.5800 ± 0.0066 | 0.3042 ± 0.0066 | 0.1888 ± 0.0028 |
| SFT | 0.5546 ± 0.0043 | 0.2998 ± 0.0021 | 0.1992 ± 0.0087 |
| Single-Head DPO | 0.6043 ± 0.0075 | 0.3055 ± 0.0100 | 0.2014 ± 0.0098 |
| DPO Soup | 0.6128 ± 0.0013 | 0.3217 ± 0.0052 | 0.2153 ± 0.0041 |
| MODPO | 0.6175 ± 0.0017 | 0.3477 ± 0.0013 | 0.2325 ± 0.0033 |
| MAH-DPO Helpful Head (Head 1) | 0.6309 ± 0.0045 | 0.3465 ± 0.0070 | 0.2239 ± 0.0098 |
| MAH-DPO Honesty Head (Head 2) | 0.6257 ± 0.0054 | 0.3516 ± 0.0078 | 0.2303 ± 0.0051 |
| MAH-DPO Truthful Head (Head 3) | 0.6257 ± 0.0010 | 0.3461 ± 0.0031 | 0.2286 ± 0.0058 |
| MAH-DPO Ensemble Head | **0.6389 ± 0.0035** | **0.3687 ± 0.0038** | **0.2478 ± 0.0074** |
| *Test-time guided decoding alignment* | | | |
| Base | 0.5750 ± 0.0107 | 0.3036 ± 0.0015 | 0.1904 ± 0.0036 |
| Helpful PRM-guided | **0.6706 ± 0.0093** | 0.4050 ± 0.0035 | 0.2791 ± 0.0023 |
| Honesty PRM-guided | 0.6448 ± 0.0050 | **0.4693 ± 0.0045** | **0.3383 ± 0.0025** |
| Truthful PRM-guided | 0.6350 ± 0.0032 | 0.4394 ± 0.0036 | 0.3296 ± 0.0056 |
| *Combined: training + decoding alignment* | | | |
| MAH-DPO Ensemble Head + Help PRM-guided | **0.7165 ± 0.0029** | 0.4554 ± 0.0028 | 0.3890 ± 0.0049 |
| MAH-DPO Ensemble Head + Honest PRM-guided | 0.6968 ± 0.0035 | **0.5196 ± 0.0016** | **0.4107 ± 0.0011** |
| MAH-DPO Ensemble Head + Truth PRM-guided | 0.6834 ± 0.0053 | 0.4872 ± 0.0038 | 0.3630 ± 0.0035 |

Table 12: Full results with standard deviations in Math.

| Method | Accuracy | Engagement |
|---|---|---|
| *Training-time alignment* | | |
| Base | 0.7107 ± 0.0090 | 0.5007 ± 0.0289 |
| SFT | 0.7300 ± 0.0060 | 0.5920 ± 0.0171 |
| Single-Head DPO | 0.7253 ± 0.0050 | 0.7160 ± 0.0257 |
| MODPO | 0.7280 ± 0.0072 | 0.7367 ± 0.0070 |
| DPO Soup | 0.7260 ± 0.0049 | 0.7353 ± 0.0075 |
| MAH-DPO Accuracy Head (Head 1) | **0.7353 ± 0.0070** | 0.8667 ± 0.0092 |
| MAH-DPO Engaging Head (Head 2) | 0.7267 ± 0.0082 | **0.8840 ± 0.0058** |
| MAH-DPO Ensemble Head | 0.7247 ± 0.0117 | 0.8733 ± 0.0069 |
| *Test-time guided decoding alignment* | | |
| Base wt normal prompt | 0.6853 ± 0.0163 | 0.5133 ± 0.0543 |
| Engaging PRM-guided wt normal prompt | 0.7013 ± 0.0352 | 0.7187 ± 0.0266 |
| Accuracy PRM-guided | 0.7633 ± 0.0050 | 0.4720 ± 0.0072 |
| Accuracy Value-guided | **0.7993 ± 0.0172** | 0.4553 ± 0.0221 |
| Base wt engaging prompt | 0.6827 ± 0.0250 | 0.7007 ± 0.0031 |
| Engaging PRM-guided wt engaging prompt | 0.7000 ± 0.0060 | **0.9033 ± 0.0050** |
| *Combined: training + decoding alignment* | | |
| MAH-DPO Ensemble Head + Accuracy Value-guided | **0.8000 ± 0.0231** | 0.8553 ± 0.0136 |
| MAH-DPO Ensemble Head + Engaging PRM-guided | 0.7107 ± 0.0114 | 0.6813 ± 0.0199 |
| MAH-DPO Ensemble Head + Engaging PRM-guided wt engaging prompt | 0.7207 ± 0.0030 | **0.9060 ± 0.0053** |

Table 13: Full results with standard deviations in Socratic Mind.

| Method | Accuracy | Engagement |
|---|---|---|
| *Training-time alignment* | | |
| Base | 0.6560 ± 0.0035 | 0.3220 ± 0.0382 |
| SFT | 0.6793 ± 0.0081 | 0.3473 ± 0.0042 |
| Single-Head DPO | 0.7040 ± 0.0053 | 0.4460 ± 0.0129 |
| MODPO | **0.7047 ± 0.0117** | 0.3600 ± 0.0122 |
| MAH-DPO Accuracy Head (Head 1) | 0.7007 ± 0.0257 | 0.4447 ± 0.0012 |
| MAH-DPO Engaging Head (Head 2) | 0.6953 ± 0.0081 | 0.4480 ± 0.0231 |
| MAH-DPO Ensemble Head | 0.6893 ± 0.0070 | **0.4513 ± 0.0127** |
| *Test-time guided decoding alignment* | | |
| Base | 0.6367 ± 0.0351 | 0.3407 ± 0.0122 |
| Accuracy PRM-guided | **0.7127 ± 0.0170** | 0.2660 ± 0.0171 |
| Engaging PRM-guided | 0.6507 ± 0.0110 | **0.4663 ± 0.0110** |
| *Combined: training + decoding alignment* | | |
| MAH-DPO Ensemble Head + Accuracy PRM-guided | **0.6659 ± 0.0210** | 0.3849 ± 0.0140 |
| MAH-DPO Ensemble Head + Engaging PRM-guided | 0.6514 ± 0.0131 | **0.5149 ± 0.0152** |

Table 14: Full results of varying head weights with standard deviations in Math.

| Weight Combination | Accuracy | Engagement |
|---|---|---|
| MAH-DPO (Accuracy head, 1.0, 0.0) | **0.7353 ± 0.0070** | 0.8667 ± 0.0092 |
| MAH-DPO (0.75, 0.25) | 0.7347 ± 0.0145 | 0.8640 ± 0.0087 |
| MAH-DPO (0.5, 0.5) | 0.7247 ± 0.0117 | 0.8733 ± 0.0069 |
| MAH-DPO (0.25, 0.75) | 0.7193 ± 0.0175 | 0.8767 ± 0.0110 |
| MAH-DPO (Engagement head, 0.0, 1.0) | 0.7267 ± 0.0082 | **0.8840 ± 0.0058** |

Table 15: Full results of varying head weights with standard deviations in Human Values.

| Weight Combination | Help | Honest | Truth |
|---|---|---|---|
| MAH-DPO (Help head, 1.0, 0.0, 0.0) | 0.6309 ± 0.0045 | 0.3465 ± 0.0070 | 0.2239 ± 0.0098 |
| MAH-DPO (0.5, 0.5, 0.0) | **0.6406 ± 0.0075** | **0.3692 ± 0.0067** | 0.2455 ± 0.009 |
| MAH-DPO (Honesty head, 0.0, 1.0, 0.0) | 0.6257 ± 0.0054 | 0.3516 ± 0.0078 | 0.2303 ± 0.0051 |
| MAH-DPO (1/3, 1/3, 1/3) | 0.6389 ± 0.0035 | **0.3687 ± 0.0038** | **0.2478 ± 0.0074** |
| MAH-DPO (0.0, 0.5, 0.5) | 0.6326 ± 0.0069 | 0.3650 ± 0.0060 | 0.2422 ± 0.0010 |
| MAH-DPO (Truth head, 0.0, 0.0, 1.0) | 0.6257 ± 0.0010 | 0.3461 ± 0.0031 | 0.2286 ± 0.0058 |
| MAH-DPO (0.5, 0.0, 0.5) | 0.6366 ± 0.0022 | 0.3645 ± 0.0085 | 0.2425 ± 0.0020 |

# G    MULTI-ACTION-HEAD DPO ABLATIONS

## G.1    PERFORMANCE UNDER CONFLICTING OBJECTIVES IN HUMAN VALUES

To stress test human values alignment under conflicting objectives, we construct a challenging subset of 7,500 preference pairs where preferences for helpfulness and honesty disagree. Each pair either selects the more helpful response while rejecting the more honest alternative, or selects the more honest response while rejecting the more helpful alternative. This subset represents an extreme case of objective conflict where a single response cannot satisfy both criteria simultaneously.

Table 16 summarizes performance on this conflict focused benchmark. Single Head DPO must route both objectives through a single action head, which forces contradictory preference signals into a shared set of parameters and leads to severe degradation, especially on honesty. In contrast, MAH DPO maintains separate gradient pathways per objective, each head optimizes its own objective specific DPO loss while the shared backbone learns cross objective representations. This architecture naturally handles conflicting preferences by learning distinct policies per head rather than averaging away the signal, achieving the strongest honesty performance while matching the best helpfulness score among baselines.

Compared with scalarization based MODPO and parameter merging based DPO Soup, MAH-DPO attains comparable or superior results across both objectives without training separate models or performing post hoc merging. The unified training framework with specialized heads therefore provides gradient isolation for conflicting preferences together with inference time flexibility, while keeping a single shared model that can be steered across objectives.

Table 16: Performance on conflicting Human Values preference subset where helpfulness and honesty preferences disagree.

| Method | Help | Honest |
|---|---|---|
| Base | 0.5800 ± 0.0066 | 0.3042 ± 0.0066 |
| SFT | 0.5546 ± 0.0043 | 0.2998 ± 0.0021 |
| Single Head DPO | 0.3397 ± 0.0084 | 0.0734 ± 0.0023 |
| MODPO | 0.6125 ± 0.0037 | 0.3386 ± 0.0028 |
| DPO Soup | 0.6071 ± 0.0042 | 0.3495 ± 0.0014 |
| **MAH-DPO** | **0.6123 ± 0.0095** | **0.3527 ± 0.0104** |

## G.2    UNIFIED MAH DPO SCALING ACROSS OBJECTIVES

Table 17: Unified MAH DPO scaling results across math and human value objectives.

| Method | Math Accuracy | Math Engagement | Help | Honest | Truth |
|---|---|---|---|---|---|
| Base | 0.7107 ± 0.0090 | 0.5007 ± 0.0289 | 0.6466 ± 0.0019 | 0.4455 ± 0.0047 | 0.3279 ± 0.0029 |
| **MAH DPO** | **0.7247 ± 0.0130** | **0.8593 ± 0.0196** | **0.6528 ± 0.0019** | **0.4516 ± 0.0020** | **0.3476 ± 0.0018** |

To study how the method scales with more objectives, we train a unified five head MAH DPO model that jointly optimizes all math and human values objectives. As shown in Table 17, the five head model improves over the base Qwen2.5 7B Instruct on every dimension, with especially strong gains on math engagement, instead of exhibiting the expected tradeoffs from negative transfer. This pattern indicates that the shared backbone captures reusable structure while the separate heads prevent direct interference. Although training with ten or more objectives is left for future work, the absence of negative transfer across these five heterogeneous heads, together with the modular backbone plus head design, suggests that the approach can scale to additional objectives beyond the ones considered here.

## H  EVALUATION ROBUSTNESS AND CROSS MODEL VALIDATION

The evaluation pipeline uses different model families for labeling and judgment across domains. In Human Values, training data are labeled by an LLM and the reward models are trained on top of Llama-3.1-8B-Instruct. In Socratic Mind, GPT-4o provides engagement labels for student surveys while PRM evaluation relies on models trained from Qwen2.5-7B-Instruct. Only the Math domain uses Qwen2.5-70B-Instruct for both preference labeling and engagement evaluation, which introduces potential circularity between training signals and test metrics.

To assess robustness under an independent judge, cross model validation is performed with GPT-4o as evaluator on the Math engagement dimension, covering both training time MAH-DPO alignment and test time PRM guided decoding. For each of 500 math problems, paired solutions are collected from the base model and from the aligned model, then GPT-4o is asked to rate engagement quality under the same instructions used in the main experiments. Both the original Qwen based judge and GPT-4o evaluate all conditions independently across three random seed runs per method.

The GPT-4o based evaluation supports the original conclusions and matches their direction and approximate magnitude. For training time MAH-DPO, 96.8% of aligned solutions are rated engaging versus 84.2% for the base model, a gain of 12.6 percentage points. For test time PRM-guided decoding, 89.87% of aligned solutions are rated engaging versus 81.13% for the base model, a gain of 8.74 percentage points. These cross judge results indicate that the observed engagement improvements are not specific to the Qwen based judge used for Math labels; both GPT-4o and the Qwen judge agree on the ranking of methods and show that MAH-DPO and PRM-guided decoding yield clear improvements over the base model. This reduces concerns about circular bias between training and evaluation and suggests that the engagement findings are stable across model families and across both training-time and test-time alignment mechanisms.

# I SOCRATIC MIND DATA SAMPLE

---

**Homework Bedtime Python Conversation Example**

**Assistant:** Consider the following Python code:

```python
homework_complete = True
time_left = 20

if homework_complete:
    print("Get ready for bed")
elif time_left < 30:
    print("Finish homework")
else:
    print("Work for 1 hour")
```

Explain what this code does. Describe each conditional statement and its purpose.

**User:** This code uses an if-elif-else structure to check different conditions and execute specific actions based on those conditions. Let's break it down:
1. `if homework_complete:` Condition: Checks if the variable homework_complete is True.
Purpose: If homework_complete is True (meaning the homework is finished), it will execute the first action (`print("Get ready for bed")`).
Outcome: If this condition is true, it prints "Get ready for bed" and skips the other conditions.
2. `elif time_left < 30:` Condition: This is checked only if the first if condition was false (meaning homework_complete is False). Now, it checks if time_left is less than 30.
Purpose: If there is less than 30 minutes left (time_left ¡ 30), it prints "Finish homework" because it suggests there's just enough time to finish the homework before heading to bed.
Outcome: This will be executed if homework_complete is False and time_left is less than 30.
3. `else:` Condition: This block is executed if both previous conditions are false (i.e., if homework_complete is False and time_left is not less than 30).
Purpose: If neither of the previous conditions is true, it prints "Work for 1 hour", implying that there is enough time (30 or more minutes) left to work on something else before going to bed.
Outcome: This is the default action that runs when none of the earlier conditions hold true.

---

## Homework Bedtime Python Conversation Example (Continued)

**Assistant:** Excellent explanation! You've clearly understood the if-elif-else structure and how the conditions are evaluated sequentially. You correctly identified that once a condition is met, the corresponding block executes and the rest are skipped.

I particularly like how you explained the purpose of each condition in context - showing you understand not just the syntax but the logic behind the code.

Now, let's move to the next question:

Modify the code snippet from the previous part so that it includes a new condition to check if `time_left` is exactly 30 minutes, printing "Almost done" in that case. Explain your modification and its impact on the code's flow.

**User:**

```
homework_complete = True
time_left = 20

if homework_complete:
    print("Get ready for bed")
elif time_left == 30:
    print("Almost done")
elif time_left < 30:
    print("Finish homework")
else:
    print("Work for 1 hour")
```

**Explanation of the Modification:** New condition: We added a new elif statement: `elif time_left == 30:`.

Purpose: This checks if time_left is exactly 30 minutes. If it is, it will print "Almost done".

This condition is evaluated before checking if time_left is less than 30 (in the previous `elif time_left < 30`), ensuring that if time_left is exactly 30, the program will print "Almost done" instead of "Finish homework".

**Impact on the flow:**

This modification adds an additional check specifically for when time_left is exactly 30 minutes.

The flow now evaluates conditions in the following order:

First: If homework_complete is True, it prints "Get ready for bed".

Second: If time_left == 30, it prints "Almost done".

Third: If time_left ¡ 30 (and the first two conditions are false), it prints "Finish homework".

Finally: If none of the above conditions are true (e.g., homework_complete is False and time_left ¿ 30), it prints "Work for 1 hour".

**Assistant:** Perfect! You've correctly modified the code and provided an excellent explanation of your changes. Your placement of the new condition is spot on - checking for `time_left == 30` before checking if it's less than 30.

I appreciate how you walked through the entire flow of the program after your modification, showing a clear understanding of how the order of conditions affects execution. You've demonstrated a solid grasp of conditional logic and how to structure conditions to achieve the desired behavior.

**Session Summary:** You demonstrated excellent understanding of conditional statements in Python. You accurately explained the if-elif-else structure, the sequential evaluation of conditions, and how only one block executes based on the first condition that evaluates to True. When modifying the code, you correctly placed the new condition in the appropriate position and explained how this affects the program's flow. Your explanations were thorough and showed strong comprehension of control flow concepts.

**Strengths:**

- Clear understanding of conditional logic and execution flow

- Ability to modify existing code to implement new requirements

- Excellent explanation of both code functionality and purpose

**Areas for continued growth:**

- Continue practicing with more complex nested conditionals

- Explore combining conditions with logical operators (and, or, not)

