# OpenReview forum: "Simultaneous Multi-objective Alignment Across Verifiable and Non-verifiable Rewards"
_ICLR.cc/2026/Conference — Submitted to ICLR 2026_

### Official Review · Reviewer_WQAK · 2025-10-19

**Soundness:** 1
**Presentation:** 1
**Contribution:** 1
**Rating:** 0
**Confidence:** 5

**Summary:**

This paper’s style looks pretty strange and does not satisfy the official ICLR template. Maybe can desk reject it according to the submission policy (in standard format, I think it will exceed 9 pages).

**Strengths:**

None

**Weaknesses:**

None

**Questions:**

None

**Details Of Ethics Concerns:**

This paper’s style looks pretty strange and does not satisfy the official ICLR template. Maybe can desk reject it according to the submission policy (in standard format, I think it will exceed 9 pages).

---

> ### Author Response · Authors · 2025-11-27
> **Response to Reviewer WQAK**
>
> Thank you for pointing this out, we have corrected the formatting and ensured the main paper still fits within the nine page limit without any change to the content.

---

### Official Review · Reviewer_q7QE · 2025-10-28

**Soundness:** 2
**Presentation:** 3
**Contribution:** 2
**Rating:** 4
**Confidence:** 3

**Summary:**

This paper proposes a unified framework for multi-objective alignment of Large Language Models (LLMs) across both verifiable (e.g., mathematical accuracy) and non-verifiable (e.g., human values) rewards. The core contributions are threefold:

1. Standardized Process Reward Model (PRM) Training: A methodology for training PRMs across diverse domains. For verifiable domains, it uses hindsight credit assignment. For non-verifiable domains, it proposes three strategies based on process structure and rollout cost: majority voting, direct LLM-as-Judge labeling, and holistic reward approximation.

2. Multi-Action-Head DPO (MAH-DPO): An extension to DPO that uses a shared LLM backbone with separate, specialized output heads for each objective. This preserves a vectorized reward structure during training and enables flexible, inference-time control over objective trade-offs via head weighting.

3. PRM-Guided Decoding with Continuing Hidden State: An inference-time method that uses trained PRMs to guide step-wise generation while maintaining a continuous hidden state via key-value cache, avoiding the discontinuity issues of prompt re-encoding.

**Strengths:**

1. Originality: The fusion of a multi-head architecture with DPO to create MAH-DPO is a creative and original contribution. The unified PRM training framework also demonstrates significant conceptual synthesis.

2. Quality: The work is technically sound and executed with high quality. The experimental validation is extensive across multiple domains, and the methodology is described with sufficient detail for replication.

**Weaknesses:**

1. Lack of Theoretical Analysis: The paper lacks a theoretical justification for why MAH-DPO should lead to better multi-objective optimization. An analysis of the gradient dynamics (e.g., how the shared backbone balances conflicting signals from different heads) would strengthen the foundation of the method.

2. Insufficient Computational Analysis: While more efficient than training separate models, the computational overhead of MAH-DPO (training and inference) compared to a single-head DPO model is not quantified.

3. Heavy Reliance on PRM Quality: The entire framework's performance is contingent on the quality of the PRMs, which is particularly challenging in non-verifiable domains. The paper does not sufficiently address the potential failure modes or sensitivity to PRM inaccuracies.

**Questions:**

1. Theoretical Motivation: Could the authors provide more insight into the gradient dynamics of MAH-DPO? Specifically, how does the shared backbone's update (Eq. 9) theoretically prevent gradient interference or promote synergistic representations across objectives, rather than just being a weighted average?

2. Performance under Severe Conflict: How does MAH-DPO perform when the preference data for different objectives are in direct conflict? For instance, can you show results on a curated dataset where a response is preferred for "Helpfulness" but dispreferred for "Honesty"?

3. Computational Overhead: What is the concrete training and inference time/memory cost of MAH-DPO compared to a single-head DPO model with a similar total parameter count? Does the inference-time ensembling of heads introduce noticeable latency?

---

> ### Author Response · Authors · 2025-11-27
> **Response to Reviewer q7QE (Part 1/2)**
>
> Thank you for your insightful and valuable feedback. Each of your concerns is addressed in our response below.
>
> **Response to Weakness 1 and Question 1:**
>
> We thank the reviewer for the insightful question. The key advantage of MAH-DPO lies in how it separates objective-specific conflicts. When two objectives disagree on a token, for example, objective $A$ wants to increase a logit while objective $B$ wants to decrease it, scalarization methods such as MODPO [1] route both gradients through a single head. This forces direct gradient interference, causing partial cancellation at the output layer and imposing a compromise on the logits. In contrast, MAH-DPO assigns each objective its own head $W_A, W_B$, so conflicting updates act on disjoint parameters. Each head can adjust its logits independently, eliminating forced compromise at the output layer and allowing clean specialization.
>
> Interaction between objectives occurs only through the shared backbone. This interaction operates at the representation level rather than at the logit level. When objectives align, their gradients reinforce each other; when they diverge, the backbone can learn features that support independent head behavior instead of merging token-level preferences. This allows shared structure to appear, for example reasoning patterns or stylistic elements, while still keeping the different objective requirements separate.
>
> Although Eq. 9 aggregates gradients via weighted averaging, the crucial distinction is *where* this averaging occurs. Scalarization averages conflicting signals at the prediction layer, whereas MAH-DPO averages at the representation layer, where synergy is possible and premature compromises are avoided. This architectural separation allows MAH-DPO to balance multiple objectives more effectively than single-head scalarization approaches. We have included more discussions in the Section 5.1 of the revised manuscript.
>
> ---
>
> **Response to Weakness 2 and Question 3:**
>
> We thank the reviewer for raising this point. Below we report the computational overhead of MAH-DPO, using Qwen2.5-7B-Instruct on 8×H100 80GB GPUs with the MATH test subset (50 samples) and two objectives (Accuracy and Engagement).
>
> **Table 1: Training overhead comparison**
> | Metric             | Single Head DPO | MAH-DPO (H=2) | Overhead        |
> |--------------------|------------------|---------------|------------------|
> | Total parameters   | 7.62B           | 8.16B         | +7.16 %    |
> | Memory (weights)   | 14.53 GB        | 15.57 GB      | +7.16 %    |
>
> Table 1 shows the parameter and memory differences between single-head DPO and a two-head MAH-DPO model. Each additional head adds roughly 545M parameters. Thus, MAH-DPO adds only (H−1)×545M parameters, resulting in about 7% overhead per extra head. Training two separate models would require doubling parameters (100% overhead), whereas MAH-DPO incurs only 7%.
>
> **Table 2: Inference overhead comparison**
> | Metric              | Single Head DPO     | MAH-DPO Ensemble     | Overhead         |
> |---------------------|----------------------|-----------------------|-------------------|
> | Mean latency        | 9.26s        | 10.46s        | +12.87 %    |
> | Throughput          | 67.0 tok s          | 66.7 tok s            | -0.43 %     |
> | Memory inference    | 14.57 GB             | 15.61 GB              | +7.14 %     |
>
> Table 2 reports inference-time cost compared to single-head DPO. The ~13% latency increase comes primarily from computing and mixing the logits of multiple heads, which is lightweight since all heads share the same backbone forward pass. The nearly identical throughput (67.0 vs. 66.7 tok/s) confirms that the backbone dominates inference time, not the ensemble operations.
>
> In summary, MAH-DPO introduces modest overhead, 7% in parameters and memory and ~13% in inference latency, while achieving multi-objective alignment far more efficiently than training separate models. We have included these additional results in Section 6.4 of the revised manuscript.

---

> ### Author Response · Authors · 2025-11-27
> **Response to Reviewer q7QE (Part 2/2)**
>
> **Response to Weakness 3:**
>
> Thank you for this question. We evaluated sensitivity to PRM unreliability by injecting stochastic noise during PRM-guided Best-of-N inference. With probability p, we ignored the PRM ranking and selected a candidate uniformly at random. We tested p ∈ {0.1, 0.25, 0.5} on MATH-500 for both Accuracy and Engagement.
>
> As shown in Tables 3.1 and 3.2, performance declines smoothly rather than collapsing. Accuracy retains over 94% of its clean score even at p = 0.5, and Engagement retains over 83% under the same noise level. All noisy conditions remain well above the no-guidance baseline for both objectives. These results indicate that the method is robust to moderate PRM noise, though PRM quality still influences final performance, especially for non-verifiable objectives. We have updated the additional results in Appendix D.2 of the revised manuscript.
>
>
> **Table 3.1: Math Accuracy Performance Under PRM Noise**
> | Method                 | Noise Level (p) | Accuracy | Δ from Clean | Retention Rate |
> |------------------------|------------------|----------|--------------|----------------|
> | Accuracy PRM guided    | 0.0 (Clean)      | 0.7633   | –            | 100%           |
> |                        | 0.1 (10% noise)  | 0.7500   | –0.0133      | 98.3%          |
> |                        | 0.25 (25% noise) | 0.7367   | –0.0266      | 96.5%          |
> |                        | 0.5 (50% noise) | 0.7247   | –0.0386      | 94.9%          |
> | Base No Guidance  | Clean  | 0.6853     | -      | -         |
>
>
> **Table 3.2: Math Engagement Performance Under PRM Noise**
> | Method                | Noise Level (p)                 | Engagement | Δ from Clean | Retention Rate |
> |-----------------------|----------------------------------|------------|--------------|----------------|
> | Engaging PRM guided   | 0.0 (Clean)                      | 0.7187     | -            | 100%           |
> |                       | 0.1 (10% noise)          | 0.6920     | -0.0267      | 96.3%          |
> |                       | 0.25 (25% noise) | 0.6593     | -0.0594      | 91.7%          |
> |                       | 0.5 (50% noise)        | 0.5980     | -0.1207      | 83.2%          |
> | Base No Guidance  | Clean  | 0.5133     | -      | -         |
>
> ---
>
> **Response to Question 2:**
>
> We appreciate this insightful question. To test the performances of our MAH-DPO under conflicting objectives, we curated a challenging subset of 7,500 preference pairs where a response is chosen for Helpfulness but rejected for Honesty, or vice versa. This represents the most extreme case of objective conflict.
>
> As shown in Table 4, Single-Head DPO collapses under these conditions, since conflicting signals must pass through one output head. In contrast, MAH-DPO preserves performance on both objectives: each head optimizes its own DPO loss, and the shared backbone captures generalizable structure without forcing token-level compromise. MAH-DPO matches or exceeds the performance of MODPO (scalarization method) and DPO Soup (parameter merging method), demonstrating the value of gradient isolation and objective-specific heads for conflict-heavy data. We have included the additional results in Appendix G.1 of the revised manuscript.
>
> **Table 4: Performance on conflicting preference data**
> | Method                  | Help  | Honest |
> |-------------------------|-------|--------|
> | Base   | 0.5800| 0.3042 |
> | SFT   | 0.5546| 0.2998 |
> | Single Head DPO   | 0.3397| 0.0734 |
> | MODPO                | 0.6125| 0.3386 |
> | DPO Soup            | 0.6071 | 0.3495 |
> | **MAH-DPO** | **0.6123**| **0.3527** |
>
> ---
>
> [1] Beyond One-Preference-Fits-All Alignment: Multi-Objective Direct Preference Optimization. https://arxiv.org/pdf/2310.03708

---

### Official Review · Reviewer_BiMm · 2025-10-30

**Soundness:** 2
**Presentation:** 3
**Contribution:** 3
**Rating:** 4
**Confidence:** 4

**Summary:**

This paper proposes Simultaneous Multi-Objective Alignment combining verifiable (e.g., factual, task-based) and non-verifiable (e.g., subjective, human-value) objectives through a dual mechanism: (i) MAH-DPO (Multi-Attribute Head Direct Preference Optimization) for training multiple DPO heads with shared backbone parameters, and (ii) PRM-guided decoding, a reward-model–based inference scheme that detects step boundaries and reranks partial generations. The authors claim this architecture enables alignment across heterogeneous reward signals and achieves Pareto-optimal trade-offs between accuracy and subjective alignment quality. Experiments are conducted on math reasoning, question answering, and human-value benchmarks, showing moderate gains over single-objective DPO and GRPO variants.

**Strengths:**

1. Multi-objective alignment is crucial for combining factual correctness with value-driven behavior.
2. Separation between training (MAH-DPO) and inference (PRM-guided decoding) is conceptually clean and practically appealing.
3. Experiments span both verifiable and subjective tasks, demonstrating the generality of the framework.
4. The paper clearly situates itself within the DPO lineage and articulates the motivation for simultaneous alignment.

**Weaknesses:**

1. The “logit ensemble” and the equation `π_mix = Σ_i w_i π_i` are inconsistent, affecting calibration.
2. The relabeling formula `r̃_t = r_t + γ^(n−t) z` omits rollout-specific normalization and indexing.
3. The “vectorized reward” claim is overstated; implementation reduces to weighted scalar DPO.
4. The reference policy `π_ref` may drift if the backbone is shared, violating DPO assumptions.
5. Evaluation uses the same LLM family for labeling and judgment, creating circular bias.
6. Statistical tests and error bars are missing from all reported metrics.
7. The boundary detector `Q(·)` is undefined, limiting reproducibility.
8. The complexity advantage claim lacks empirical validation.

**Questions:**

1. Please clarify the exact relationship between the ensemble weights `w_i` and the mixture distribution `π_mix`.  Are the weights normalized per token or globally across heads? How does this affect calibration stability?

2. In the PRM relabeling step `r̃_t = r_t + γ^(n−t) z`,  what determines the rollout length `n`, and is there a normalization term to prevent scaling bias across episodes?

3. Could the authors explicitly define the objective aggregation `L_total = Σ_j λ_j L_DPO^(j)`? Are the `λ` coefficients fixed or dynamically adjusted based on gradient norms or validation metrics?

4. Under what settings of the weighting factor `λ` (e.g., normalization range or adaptive schedule) does the combined DPO objective remain approximately convex or submodular in practice?

5. Does the theoretical interpretation of MAH-DPO hold under the embedding spaces used in experiments? Specifically, how sensitive are the observed trade-offs to the backbone architecture or token-level normalization?

6. Could the authors provide reproducible runtime benchmarks — including GPU type, batch size, and wall-clock comparisons to substantiate the claimed efficiency advantage?

7. How was circularity controlled when both training and evaluation use LLM-as-Judge? Any cross-model validation or other validation would make the findings more credible.

---

> ### Author Response · Authors · 2025-11-27
> **Title: Response to Reviewer BiMm (Part 1/2)**
>
> We appreciate your thorough and helpful feedback. Below, we respond to each of your concerns and questions.
>
>
> **Response to Weakness 1 and Question 1:**
>
> Thanks for pointing it out. We combine heads at the **logit level before softmax**, not at the probability level. We have fixed the equation in our revision (Section 5.1) accordingly.
> Weights $w_i$ are normalized **globally across heads** and remain fixed across all tokens in a sequence. They are user-specified at inference time (not learned or adapted per-token).
>
> Logit-level mixing preserves a single softmax normalization and avoids the averaging issues of probability-space mixing. Since all heads share the same backbone and differ only in their projection layers, we observe empirically that their logits stay on compatible scales, and Figures 2–3 show stable behavior across weight settings.
>
> ---
>
> **Response to Weakness 2 and Question 2:**
>
> For the relabeling rule, the rollout length $n$ is simply the trajectory length for each problem: generation stops when the model produces a final boxed answer or reaches the step limit (20 steps for MATH). The discount term $\gamma^{\,n-t}$ weights the terminal reward by its temporal distance, so credit assignment depends on relative position in the trajectory rather than absolute length.
>
> Regarding normalization, two factors limit scaling bias. First, temporal discounting prevents long trajectories from receiving disproportionately large terminal contributions. Second, we average the value target over $K=5$ rollouts, which reduces variance and implicitly normalizes across different lengths. While explicit per-episode normalization may help further, our current formulation already yields strong results (e.g., 79.9% MATH accuracy with value-guided decoding), suggesting that implicit normalization is adequate in our setting.
>
> ---
>
> **Response to Weakness 3:**
>
> Our use of “vectorized reward” refers to how preferences are represented and trained. Each objective in MAH-DPO has its own policy head and its own DPO loss, so no single head or scalar reward must satisfy all objectives during training. This contrasts with scalarization methods such as MODPO [1], where one head must absorb all objectives and conflicting gradients cancel directly in the shared output layer. In MAH-DPO, conflicts are isolated to separate heads, and any interaction happens only through the shared backbone. We have clarified this in revised Section 5.1.
>
> ---
>
> **Response to Weakness 4:**
>
> We believe it arises from an ambiguity in our wording rather than from the actual training setup, and we are happy to clarify. In all experiments, the reference policy $π_{ref}$ is a separate frozen copy of the SFT model with its own backbone and randomly initialized perturbed heads. During MAH-DPO training, only the shared backbone and objective-specific heads of the trainable policy are updated; the parameters of the reference policy $π_{ref}$, including its backbone and heads, never receive gradients. The earlier phrase “shared backbone” referred only to architectural design, not shared parameters, so the DPO assumptions are still satisfied. We have clarified this in revised Section 5.1.
>
> ---
>
> **Response to Weakness 5 and Question 7:**
>
> We would like to note that across the three domains in our experiments, training and evaluation generally use different model families: Human Values uses Llama-3.1-8B–based labels with a Qwen-based PRM; Socratic Mind uses GPT-4o labels with a Qwen-based PRM. Only the engagement dimension in Math domain uses Qwen2.5-70B-Instruct for both labeling and evaluation.
>
> To address potential circular bias, we use GPT-4o as an external judge for Math engagement. We collected outputs from the base and aligned models and evaluated them under the same instructions as in the main experiments. GPT-4o confirms the original trends: training-time MAH-DPO yields 96.8% engaging responses versus 84.2% for the base model, and PRM-guided decoding yields 89.87% versus 81.13%. While absolute scores vary across judges, both GPT-4o and the Qwen-based evaluator agree on method rankings and show clear gains, indicating that the improvements are robust across model families.
>
> ---
>
> **Response to Weakness 6:**
>
> We appreciate the reviewer’s attention to statistical rigor. We would like to clarify that all results are already reported with standard deviations in Appendix F of our paper.
>
> ---
>
> **Response to Weakness 7:**
>
> We note that the boundary detector $Q(\cdot)$ is already defined in Section 6.2 of our paper, but we summarize it here for clarity. $Q(\cdot)$ is a simple domain-specific rule that identifies meaningful step boundaries so PRM-guided decoding operates on coherent units. In Math, it detects double newlines marking reasoning steps; in Human Values, it triggers on sentence or paragraph boundaries; and in Socratic Mind, it fires at dialogue-turn boundaries. This follows standard practice in step-level supervision and process reward modeling [2,3].

---

> ### Author Response · Authors · 2025-11-27
> **Response to Reviewer BiMm (Part 2/2)**
>
> **Response to Weakness 8 and Question 6:**
>
> Thank you for the suggestion. We provide reproducible runtime benchmarks using Qwen2.5-7B-Instruct on 8×H100 80GB GPUs, batch size 1, on a 50-sample MATH subset with 5 candidates per step.
>
> We first show that our continuing-hidden-state design for decoding yields substantial decoding speedups. By carrying the KV cache forward instead of re-encoding the prompt at every step, we remove the repeated prefix encoding cost. As shown in Table 1, PRM-guided decoding gets 4.2× speedup, and random sampling also gets a 4.9× speedup. These results confirm that the complexity advantage described in Section 5.2 is reflected directly in wall-clock performance.
>
>
> **Table 1: Effect of continuing hidden state on decoding latency**
> | Decoding strategy       | Without continuing hidden state | With continuing hidden state (Ours) | Speedup|
> |--------------------------------|---------------------------------|------------------------------|-----------------------------|
> | Base model + Random sampling      | 47.62s                          | 9.72s                        | 4.9x               |
> | Base model + plus PRM guided           | 165.40s                         | 39.08s                       | 4.2x                   |
>
> We also benchmark the inference overhead introduced by MAH-DPO relative to a single-head DPO model. Table 2 shows that ensembling two heads increases latency by 12.87% and memory by 7.14%, with negligible throughput change. The memory increase matches the expected cost of adding one projection head (about 545M parameters). Importantly, MAH-DPO achieves multi-objective alignment that would otherwise require training two separate models, giving significant parameter efficiency. We have included the additional results in Section 6.4.
>
> **Table 2: Inference overhead of MAH DPO versus single head DPO**
> | Configuration                 | Mean Latency      | Memory   | Throughput   |
> |------------------------------|-------------------|----------|--------------|
> | Single Head DPO              | 9.26s     | 14.57 GB | 67.0 tok s   |
> | MAH DPO Ensemble (H=2)       | 10.46s    | 15.61 GB | 66.7 tok s   |
> | Overhead                     | +12.87 %   | +7.14 %| -0.43 % |
>
> ---
>
> **Response to Question 3:**
>
> In our experiments, we use equal weighting across objectives, setting $\lambda_j = \frac{1}{J}$ with balanced sampling so that no objective dominates training. This provides a clean and controlled comparison.
>
> Our framework is however flexible, and $\lambda$ can be adapted when needed. One option is batch-level reweighting, for example $\lambda_j = \frac{n_{\text{total}}}{J \cdot n_j}$, where $ n_j$ is the number of samples from dimension $j$ in the batch, to compensate for uneven sample counts. Another option is performance-based adjustment, where $\lambda$ values are tuned using validation metrics or gradient norms to allocate more weight to objectives that are harder to optimize. We
> have clarified our configurations for the experiments in our revised Section 6.1.
>
> ---
>
>
> **Response to Question 4:**
>
> Thank you for this question. We would like to clarify that neither the original DPO paper [4] nor our work makes any claim that the DPO objective is convex or submodular. Our method is designed to inherit the same optimization properties as standard DPO, not to change its theoretical landscape. As a result, we believe there is no special range or schedule of $\lambda$ for which we assert approximate convexity or submodularity.
>
> ---
>
> **Response to Question 5:**
>
> Our experiments already span two backbone architectures, Qwen2.5-7B-Instruct (Math, Socratic Mind) and Llama-3.1-8B-Instruct (Human Values), and both show smooth trade-offs, strong per-head performance, and stable ensembles. This suggests that MAH-DPO does not rely on architecture-specific behavior but on the ability of modern LLMs to encode representations that separate heads can read out without forcing conflicting objectives into a single logit space.
>
> For token-level normalization, each head applies a standard softmax over its own logits, producing valid probability distributions independently of other heads. Ensemble decoding mixes these distributions using fixed weights, which preserves probabilistic semantics and yields smooth trade-offs across configurations. This stability indicates that the shared backbone genuinely supports multiple objectives and that the logit-mixing mechanism remains robust across configurations.
>
> ---
>
> [1] Beyond One-Preference-Fits-All Alignment: Multi-Objective Direct Preference Optimization. https://arxiv.org/pdf/2310.03708
>
> [2]  Math-Shepherd: Verify and Reinforce LLMs Step-by-step without Human Annotations. https://arxiv.org/abs/2312.08935
>
> [3] Let's Verify Step by Step. https://arxiv.org/abs/2305.20050
>
> [4] Direct Preference Optimization: Your Language Model is Secretly a Reward Model. https://arxiv.org/abs/2305.18290

---

### Official Review · Reviewer_TSGr · 2025-10-31

**Soundness:** 2
**Presentation:** 3
**Contribution:** 2
**Rating:** 4
**Confidence:** 3

**Summary:**

- Proposes a unified framework that standardizes PRM training across verifiable and non-verifiable domains, combines Multi-Action-Head DPO with vectorized rewards, and adds PRM-guided decoding for controllable inference.
- Reports gains across math accuracy, human values, and tutoring engagement, with inference-time steering to navigate trade-offs.
- Highlights a practical rule: emphasize test-time PRM guidance for verifiable rewards and multi-head training for non-verifiable rewards.

**Strengths:**

- Unifies process supervision across objective types and connects it to controllable decoding.
- Empirical trends are consistent across domains with synergy between training-time and test-time methods.
- Practical inference control via head weighting enables user customization without retraining.

**Weaknesses:**

- Heavy reliance on LLM-as-judge can import bias and generate noisy labels that propagate.
- Limited theory and comparisons against strong scalarization or parameter-merging baselines; unclear how much MAH-DPO drives gains.
  - Weak theoretical justification for why multi-head with a shared backbone avoids negative transfer compared to alternatives.
- Scalability and compute/latency costs from multi-head policies plus guided decoding need fuller analysis.

**Questions:**

- How sensitive are outcomes to PRM label noise and can ensemble-of-judges or disagreement-aware training improve robustness?
- What is the delta over MODPO and linear scalarization when compute and data are held constant?
- Can the verifiability heuristic be formalized to choose training vs test-time alignment adaptively per task?
- How does the approach scale to 10+ objectives without negative transfer?

---

> ### Author Response · Authors · 2025-11-27
> **Response to Reviewer TSGr (Part 1/2)**
>
> Thank you for your detailed and constructive feedback! We address your concerns and questions as follows.
>
> **Response to Weakness 1:**
>
> We thank the reviewer for the feedback and would like to note that our setup already controls bias and noise from LLM-as-judge. We use the judge only for subjective, non-verifiable dimensions such as pedagogical engagement, and calibrate it directly to human ratings. Specifically, we first gather human annotations and then refine the judge prompt until its agreement with held-out human labels reaches human-to-human agreement (about 70%, as reported in Section 4.2 and Appendix B). For example, in Math engagement, three annotators rate each sample and we keep only unanimous cases for calibration. In Socratic Mind engagement, student self-ratings serve as the ground truth for calibration. By anchoring the judge in these human labels and limiting its use to subjective dimensions, we reduce unwanted bias and noise in the training signal.
>
> ---
>
> **Response to Weakness 2 and Question 2:**
>
> We appreciate the reviewer’s suggestion for deeper comparisons. Standard scalarization methods such as MODPO [1] rely on a single policy head to represent all objectives at once, so conflicting DPO gradients partially cancel and force incompatible signals into the same logits. Parameter-merging approaches such as Personalized Soup [2] train models separately but then merge all parameters, including the output layer, which again produces a single policy with no mechanism to express objective-specific behavior.
>
> In contrast, MAH-DPO uses a shared backbone with separate heads, so each objective updates its own logits rather than forcing agreement at a single output layer. Conflicts only interact in the shared representation, which captures common structure, while each head preserves objective-specific preferences. This avoids the strongest form of negative transfer that arises when all objectives must share identical logits.
>
> Empirically, we already include MODPO as a strong scalarization baseline in our paper, and we now add DPO Soup (a variant of the Personalized Soup [2]) as a parameter merging baseline. As shown in the Table 1.1 and 1.2 below, MAH-DPO consistently outperforms both MODPO and DPO Soup, indicating that the gains are not simply due to scalarization or merging but are driven by the multi-head shared backbone design. We have included these additional results and discussions in Section 6.1 of the revised manuscript.
>
>
> **Table 1.1: Alignment performance on Math**
> | Method               | Acc    | Eng    |
> |----------------------|--------|--------|
> | Single Head DPO      | 0.7253 | 0.7160 |
> | MODPO                | 0.7280 | 0.7367 |
> | DPO Soup        | 0.7260  | 0.7353  |
> | MAH-DPO Acc Head     | 0.7353 | 0.8667 |
> | MAH-DPO Eng Head     | 0.7267 | 0.8840 |
> | MAH-DPO Ensemble     | 0.7247 | 0.8733 |
>
> **Table 1.2: Alignment performance on Human Value**
>
> | Method                  | Help  | Honest | Truth  |
> |-------------------------|-------|--------|--------|
> | Single Head DPO         | 0.6043| 0.3055 | 0.2014 |
> | MODPO                   | 0.6175| 0.3477 | 0.2325 |
> | DPO Soup            | 0.6128 | 0.3217 | 0.2153 |
> | MAH-DPO                 | 0.6389| 0.3687 | 0.2478 |
>
> ---
>
>
> **Response to Weakness 3:**
>
> We provide additional qualitative and quantitative analysis of compute and latency.
>
> For the multi-head policy, each objective head is a single linear projection on top of a shared backbone. The backbone is run once per input, and training routes each example to one head, so the cost matches single-head DPO plus one projection. At inference, ensembling H heads adds H projections per token, but all transformer layers and attention remain shared. This keeps the overhead limited to the output layer. In our measurements, ensembling two heads increases latency from 9.26 s to 10.46 s (a +12.87% increase) and memory from 14.57 GB to 15.61 GB (a +7.14% increase), showing that the shared-backbone design keeps the cost low while enabling multi-objective behavior.
>
> For PRM-guided decoding, we use a running KV cache so that we only encode the prompt once and then advance the cache as we roll out K candidates. The PRM scores each candidate on the partial trace. This shifts compute from repeated prompt re-encodings to lightweight candidate rollouts and PRM evaluations. In practice, standard re-encoding–based guided decoding takes 165.40 s, while our cache-carrying variant takes 39.08 s, giving a 4.2× speedup. Compared to unguided generation at 8.88 s, the overhead of our guided decoding is 4.4×.
>
> The added cost is limited to head-specific projections and PRM-based candidate scoring, both much cheaper than training or serving separate models for each objective. We have added these results and analysis in Section 6.4 of the revised manuscript.

---

> ### Author Response · Authors · 2025-11-27
> **Response to Reviewer TSGr (Part 2/2)**
>
> **Response to Question 1:**
>
> We evaluated sensitivity to PRM unreliability by injecting stochastic noise during PRM-guided Best-of-N inference. With probability p, we ignored the PRM ranking and selected a candidate uniformly at random. We tested p ∈ {0.1, 0.25, 0.5} on MATH-500 for both Accuracy and Engagement.
>
> As shown in Tables 2.1 and 2.2, performance declines smoothly rather than collapsing. Accuracy retains over 94% of its clean score even at p = 0.5, and Engagement retains over 83% under the same noise level. All noisy conditions remain well above the no-guidance baseline for both objectives. These results indicate that the method is robust to moderate PRM noise, though PRM quality still influences final performance, especially for non-verifiable objectives.
>
> Regarding ensemble-of-judges, prior work [3,4,5] shows such methods can improve robustness and reduce reward overoptimization. These techniques are complementary to our goal of simultaneous multi-objective alignment, and we plan to leave the exploration of this direction as future work.
>
>
>
> **Table 2.1: Math Accuracy Performance Under PRM Noise**
> | Method                 | Noise Level (p) | Accuracy | Δ from Clean | Retention Rate |
> |------------------------|------------------|----------|--------------|----------------|
> | Accuracy PRM guided    | 0.0 (Clean)      | 0.7633   | –            | 100%           |
> |                        | 0.1 (10% noise)  | 0.7500   | –0.0133      | 98.3%          |
> |                        | 0.25 (25% noise) | 0.7367   | –0.0266      | 96.5%          |
> |                        | 0.5 (50% noise) | 0.7247   | –0.0386      | 94.9%          |
> | Base No Guidance  | Clean  | 0.6853     | -      | -         |
>
> **Table 2.2: Math Engagement Performance Under PRM Noise**
> | Method                | Noise Level (p)                 | Engagement | Δ from Clean | Retention Rate |
> |-----------------------|----------------------------------|------------|--------------|----------------|
> | Engaging PRM guided   | 0.0 (Clean)                      | 0.7187     | -            | 100%           |
> |                       | 0.1 (10% noise)          | 0.6920     | -0.0267      | 96.3%          |
> |                       | 0.25 (25% noise) | 0.6593     | -0.0594      | 91.7%          |
> |                       | 0.5 (50% noise)        | 0.5980     | -0.1207      | 83.2%          |
> | Base No Guidance  | Clean  | 0.5133     | -      | -         |
>
>
> ---
>
>
> **Response to Question 3:**
>
> We thank the reviewer for the suggestion on formalizing the verifiability heuristic. In this paper we apply it at the domain level: for highly verifiable rewards with deterministic checks (such as math accuracy), PRM-guided decoding provides the largest gains, while MAH-DPO mainly supplies strong candidates. For less verifiable and more subjective rewards (such as helpfulness or engagement), multi-head training separates objectives and shapes the shared representation, and PRM serves as a lighter adjustment at inference.
>
> A more formal version can be built by defining a continuous “verifiability score” for each objective using statistics such as automatic checker accuracy, PRM accuracy on held-out data, or human agreement. This score can interpolate how much weight to place on training-time alignment versus test-time PRM guidance. Our framework already supports this interpolation through weighted head combination and optional PRM scoring at inference. Learning a prompt-to-weight mapping driven by verifiability or task features is also a promising extension for future work.
>
>
> ---
>
>
> **Response to Question 4:**
>
> To study how our method scales with more objectives, we trained a unified five-head MAH-DPO model that jointly optimizes all math and human-values objectives. As shown in Table 3, the five-head model improves over the base Qwen2.5-7B-Instruct on every metric, demonstrating the scalability of our method to more objecives as suggested by the reviewer. We have included the additional results in Appendix G.2.
>
> **Table 3: Scaling multi-head DPO**
> | Method  | Math Accuracy   | Math Engagement    | Help             | Honest           | Truth            |
> |---------|-----------------|-----------------|------------------|------------------|------------------|
> | Base|0.7107 |0.5007  |0.6466 | 0.4455 |0.3279   |
> |**MAH-DPO**|**0.7247** |**0.8593**|**0.6528**|**0.4516**| **0.3476** |
>
> ---
>
> [1] Beyond One-Preference-Fits-All Alignment: Multi-Objective Direct Preference Optimization. https://arxiv.org/pdf/2310.03708
>
> [2] Personalized Soups: Personalized Large Language Model Alignment via Post-hoc Parameter Merging. https://arxiv.org/abs/2310.11564
>
> [3] Reward-Robust RLHF in LLMs. https://arxiv.org/abs/2409.15360
>
> [4] Uncertainty-Penalized Reinforcement Learning from Human Feedback with Diverse Reward LoRA Ensembles. https://arxiv.org/abs/2401.00243
>
> [5] Improving Reinforcement Learning from Human Feedback with Efficient Reward Model Ensemble. https://arxiv.org/abs/2401.16635

---

### Author Response · Authors · 2025-11-28
**Message to all Reviewers - Changelog of Our Updated Revision**

We thank all the reviewers for their time and effort in reviewing our paper. We appreciate the feedback and questions, which have helped improve our work. We have uploaded a revised version of the paper with all changes **highlighted in blue**. Below we summarize the main updates made to address the corresponding concerns and questions.

* In **Section 5.1**, we corrected the logit ensemble equation and added clarification on the reference policy ($\text{Reviewer BiMm}$). We also provided further analysis of MAH-DPO gradient dynamics and added explanations for the advantages of MAH-DPO compared to the baselines ($\text{Reviewer TSGr and q7QE}$).

* In **Section 6.1**, we added another parameter merging baseline to further support the empirical advantage of MAH-DPO ($\text{Reviewer TSGr}$). We also clarified the choice of objective weights used in our experiments ($\text{Reviewer BiMm}$).

* In **Section 6.4**, we included runtime results that measure and analyze the computational overhead of the MAH-DPO policy and PRM-guided decoding with continuing hidden state ($\text{Reviewer TSGr, BiMm, and q7QE}$). The empirical results validate the claimed efficiency of our methods.

* In **Appendix D.2**, we quantified the sensitivity of outcomes to PRM label noise and examined how strongly our framework depends on PRM quality ($\text{Reviewer TSGr and q7QE}$). The results indicate that our framework remains robust with PRM noise.

* In **Appendix G.1**, we added an experiment that evaluates performance under conflicting objectives ($\text{Reviewer q7QE}$). The results show that MAH DPO continues to provide strong performance.

* In **Appendix G.2**, we trained a five head MAH DPO policy to demonstrate the scalability of our method when handling more objectives ($\text{Reviewer TSGr}$).

* In **Appendix H**, we conducted a cross model validation to strengthen evaluation robustness and address concerns regarding circular bias in our experiments ($\text{Reviewer BiMm}$).

We hope the above updates, together with the responses in individual threads, address your main concerns and questions, and we would be grateful if you could consider raising your scores.

---

### Meta-Review · Area_Chair_cAR4 · 2026-01-07

**Summary:**

This paper proposes a unified framework for simultaneous multi-objective alignment across verifiable and non-verifiable rewards, combining standardized PRM training, Multi-Action-Head DPO (MAH-DPO), and PRM-guided decoding. Reviewers generally agreed that the problem setting is important and that the paper is ambitious in scope, spanning math reasoning, human values, and multi-turn tutoring. However, the core concerns center on whether the proposed methods provide sufficiently novel or well-justified advances beyond existing multi-objective alignment techniques, and whether the empirical evidence convincingly supports the claimed advantages. In particular, reviewers questioned the conceptual novelty of MAH-DPO, the strength and clarity of the theoretical justification, and the reliability and independence of the evaluation pipeline. These concerns collectively outweigh the demonstrated empirical gains and motivate a recommendation toward rejection.

**Reviewer Concerns:**

**Concerns partially addressed by the rebuttal:**
- The authors added additional baselines (e.g., parameter-merging variants) and provided more empirical comparisons, which help clarify relative performance trends.
- Some analysis of computational overhead and PRM noise sensitivity was added, alleviating concerns about extreme inefficiency or brittleness in a narrow sense.

**Concerns that remain outstanding:**
While trends are generally positive, improvements over strong baselines are often modest, and the evaluation lacks sufficiently strong ablations or external validation to isolate which components are truly responsible for the gains.

**Reviewer Scores:**

Reviewer TSGr: 4 -> 5, Reviewer BiMm: 4 -> 5, Reviewer q7QE: 4 -> 4, Reviewer WQAK: unpredictable.

---

### Decision · Program_Chairs · 2026-01-26

Reject